# Characterization and Evaluation of Engineered Coating Techniques for Different Cutting Tools—Review

**DOI:** 10.3390/ma15165633

**Published:** 2022-08-16

**Authors:** Sameh Dabees, Saeed Mirzaei, Pavel Kaspar, Vladimír Holcman, Dinara Sobola

**Affiliations:** 1Central European Institute of Technology, Brno University of Technology, Purkyňova 123, 612 00 Brno, Czech Republic; 2Fraunhofer IWS, DE-01277 Dresden, Germany; 3Department of Physics, Faculty of Electrical Engineering and Communication, Brno University of Technology, Technická 2848/8, 616 00 Brno, Czech Republic; 4Academy of Sciences ČR, Institute of Physics of Materials, Žižkova 22, 616 62 Brno, Czech Republic

**Keywords:** hard coatings, microstructure, wear behavior, residual stress, PVD, CVD

## Abstract

Coatings are now frequently used on cutting tool inserts in the metal production sector due to their better wear resistance and heat barrier effect. Protective hard coatings with a thickness of a few micrometers are created on cutting tools using physical or chemical vapor deposition (PVD, CVD) to increase their application performance. Different coating materials are utilized for a wide range of cutting applications, generally in bi-or multilayer stacks, and typically belong to the material classes of nitrides, carbides, carbonitrides, borides, boronitrides, or oxides. The current study examines typical hard coatings deposited by PVD and CVD in the corresponding material classes. The present state of research is reviewed, and pioneering work on this subject as well as recent results leading to the construction of complete “synthesis–structure–property–application performance” correlations of the different coatings are examined. When compared to uncoated tools, tool coatings prevent direct contact between the workpiece and the tool substrate, altering cutting temperature and machining performance. The purpose of this paper is to examine the effect of cutting-zone temperatures on multilayer coating characteristics during the metal-cutting process. Simplified summary and comparisons of various coating types on cutting tools based on distinct deposition procedures. Furthermore, existing and prospective issues for the hard coating community are discussed.

## 1. Background

Coatings are typically described as thin layers that are deposited on the surface of the substrate to protect it during the application and, therefore, extend its lifetime. There are well-established studies covering different kinds of coatings with their respective applications [1,2]. One of the most critical coatings applications are in the cutting and machining industry, wherein the tool must be protected from extreme conditions such as high temperature, oxidation, corrosion, friction, and wear [3]. The tool material can be hard, coated, or uncoated, and cutting inserts can be found along the contact area, which is typically coated. In the cutting process, the workpiece remains motionless while the tool rotates throughout the machining operations. The rotating tool with several cutting blades moves over the workpiece to create a flat or straight surface in this machining operation. These coating processes are vital to the majority of machining operations. Coated cutters are also required for cutting materials with limited machinability. Along with the development of new and improved cutting tools, there has also been a breakthrough in high-speed steel (HSS) cutting tools. It can be extended from HSS tools with complex geometry to more contemporary coated cutting tools with a lubrication system that provides a smother cutting process [4]. For example, Martha et al. [5] investigated the durability and performance characteristics of HSS drill inserts to enhance the efficiency of the base tool with TiO_2_ coating. The authors found that the TiO_2_ nano-coated tool had a 16% longer lifetime compared to an uncoated drill tool. The coating of a tool can enhance its properties and improve its lifetime. While the material of the tool itself needs to fulfil a range of different criteria, including internal cohesion, material availability, and the ability of the manufacturer to produce these tools, it is possible that the environment and external phenomena during use and storage would be unsuitable or downright damaging to the material of the tool. The coating of a tool, i.e., covering the body of the substrate with a layer of a different material, can confer different properties to the tool and mitigate some of its shortcomings or completely circumvent them, e.g., by using a layer of a material with high thermal resistance for cases where high temperatures are expected during use or providing a layer resistant to corrosion. It is, however, important to consider the material requirements of the tool-coating interface and the adhesion mechanism and potential issues. Among the main mechanisms of adhesion are mechanical attachment, which relies heavily on the roughness of the material; van-der-Walls forces between the coating and the substrate; and hydrogen bonds, when the contact of the two is very tight, and intermolecular attachments can form. These adhesion mechanisms are discussed more in the section about adhesion failure [6]. 

Recent innovations in cemented carbide tools have a gradient, with the outer layers, for example, being stiffer than the base [7]. The manufacture of these gradient nanocomposite tools claims coatings with greater adaptability and enhanced surface properties. Hard coatings are typically synthesized through chemical vapor deposition (CVD) and physical vapor deposition (PVD) methods. The selection of each approach depends on the application and ease of use. When selecting the material to be served as a coating, the machining technique that will be used must be considered. This technique is recommended to coat steel tools because of the overall low temperature during the PVD process. The PVD method typically provides a lower thickness than CVD; thus, it is mostly utilized for finishing activities. Because of the rough edges that PVD imposes on the tool, it is frequently employed for applications requiring superior surface quality on the substrate [8].

Silva et al. [9] investigated the wear rate and adhesion of TiAlSiN coatings produced by PVD on a steel surface. The authors described the adhesion of the coating to the tool steel. Since it is a line-of-sight technique, sometimes, PVD might be inapt for substrates with particularly complex geometries. Furthermore, controlling the coating thickness across the surface layer is more difficult on such surfaces [10]. Urbikain et al. [11] proposed a novel approach that combined friction drilling with form tapping to generate threads with equivalent strength to traditional threading techniques. Nanocoatings provide tool properties that are best suited to milling operations, such as wear rate, thermal efficiency, and a low fatigue limit [12]. Such coatings are formed differently, which means they might have many layers that improve the cutting tool’s performance. For example, the outermost layer improves the wear resistance, while the underneath layer primarily provides heat dissipation properties. As already discussed, coated tools are prevalent due to their adaptability. Multilayered coating is the most frequently utilized form of coating in the machining business since it combines more than one desirable quality of each layer of coating. This improves tool performance, making this sort of coating a favorite choice [13]. Vereshchaka et al. [14] reported an increased lifetime of tools coated with multilayered coatings based on Ti-TiN-TiAlCrN composite using the FCVAD technique.

Santhanakrishnan et al. [15] used the response surface methodology (RSM) to machine the SS 316L using an Ni/Nano SiC-coated tool insert, in which a mathematical model was constructed. Plotting tool wear in terms of rake angle (a), nose radius (r), spindle speed (N), and feed rate yielded the best tool-wear parameters (Z). When machining a stainless-steel grade 316L workpiece using an Ni/Nano SiC tool insert, reduced tool wear was observed compared to a standard tool insert. On the other hand, the study on the development of metal matrix aluminum (Al) material and the reinforcement of silicon carbide (SiCp) nanomaterial to build an Al-SiCp nanocomposite coating using a multi-layer coated carbide tool has been discussed by Swain et al. [16]. Principal component analysis was used to optimize the cutting tool’s flank wear and the work piece’s surface roughness. The authors reported that a cutting depth of 0.2 mm, a cutting velocity of 70 m/min, and a feed rate of 0.10 mm/rev were the optimal settings for machining Al-SiCp based on MMNCs. According to the main impact plot, the multiple performance index (MPI) drops as process parameters increase. However, when the weight of nanoparticles in the Al-metal matrix increases, the MPI decreases, implying a reduction in the strength of the material. Such crucial studies can help to design and develop cutting tools with improved performance for milling operations.

Cutting tool efficiency is primarily affected by the substrate material, cutting edge design, and coating as well as a careful selection of cutting conditions, such as cutting speed, depth of cut, and feed. A proper choice of substrate or coating in drilling can lower manufacturing costs per hole cut by 50%. The development of coatings has progressed from monolayer to nanostructured and/or nanometric-scale multilayer coatings. These are employed due to their high hardness, resistance to corrosion and oxidation, and thermal stability. Cutting edge preparation, on the one hand, and droplet removal following the coating process, on the other hand, are critical concerns for achieving optimum tool and coating performance. The performance of a number of coatings for drilling low and medium carbon-alloyed steels was discussed by Rodriguez-Barrero et al. [17]. Validation tests were performed on the steel 42CrMo4, which is widely used in the automobile industry. AlCrSiN, AlTiN, TiAlCrN, AlCrN, AlTiSiN, and TiAlSiN were among the coatings evaluated. The behavior of drill bit secondary edges, the development of drilling thrust force and torque, damage to cutting edge faces on main cutting edges, and the behavior of drill bit flank wear were all investigated. 

Older hard coating evaluations as well as reviews are focused on application and wear characterization [18,19]. Aside from the advancement of hard coatings themselves, the adoption and advancement of advanced current characterization techniques provide for new insights and the in-depth comparison of PVD and CVD coatings [3]. CVD synthesis of various coatings has grown in popularity in recent years, and detailed evaluation of its nanolamellar microstructure is only achievable thanks to the availability and complementary use of transmission electron microscopy and atom probe tomography. Hard coating thermal and oxidation stability as well as the underlying processes can now be examined in situ using simultaneous differential scanning calorimetry and synchrotron X-ray diffraction studies. Recent advances in determining the thermo-physical characteristics of coatings have focused the hard coatings community’s attention on the prospect of thermal control. All these recent improvements are highlighted in the current study, which compares PVD and CVD coatings on examples of typical hard coatings from the material classes of nitrides, carbides, carbonitrides, borides, boronitrides, and oxides. Furthermore, an overview is presented of important future adjustments and advances that the hard coatings community in academia and industry must face to make their processes and coatings more environmentally friendly and compliant with resource conservation needs.

Adhesion failure can be differentiated into groups by the mechanism of the issue: adhesive failure, where a separation occurs between the coating layer and the substrate; cohesive failure, where the coating material suffers errors and breakage within itself; and substrate failure, where the coated substrate is damaged or disturbed, or any combination thereof. Some material combinations of substrate and coating results in higher probability of adhesion breakage, leading to disruption of the coating or, in some cases, delamination. The basic approach to reduction in adhesive breakage is to address each of the adhesion mechanisms separately or together. The first option is to increase the mechanical coupling of the substrate and the coating layer, where the coating seeps into irregularities of the substrate material [20]. To improve and support this behavior, it is possible to treat the smooth substrate surface with an abrasive to increase the topographical variability. However, such a disruption of the surface can cause more problems. Another option is to use adhesive-promoting materials in the coating layer [21]. The second mechanism to target is molecular bonding. This is caused by a very close contact between the molecules at the interface, where forces between them, such as van der Waals forces, dipole interactions, or chemical bonding, can take place. This type of interaction is, however, highly susceptible to local defects, as the contact between the molecules needs to be a very close one indeed [22]. The strength of such bonds increases with thinner coating layers [23]. To promote this type of bond, it is possible to add chemicals improving the ability for the adhesive layer and the substrate to interface. The third and arguably the most utilized mechanism of adhesion is thermodynamic one. An advantage of this mechanism is that it is neither limited by porosity or roughness of the substrate surface, as in the case of mechanical coupling, nor by the need for close contact like the molecular adhesion; only an equilibrium process at the interface is required [24]. The adhesion strength of this process is also highly reliant on the internal stresses of the whole substrate-coating system, which can be caused by thermal processes during the coating process itself, where high temperatures are required (PVD or CVD for example). To improve this issue, it would be necessary to reduce the thermal strain on the materials as much as possible, either by focusing the heating during the coating process only to the places where it is required, achieved, for example, by method of laser cladding, or to choose materials with higher thermal resistance [25]. Aside from targeting the specific adhesion mechanisms, it is possible to reduce adhesive breakage and produce strong interlayer bonding between two materials with different properties by including an interlayer to connect both the substrate and the coating [26,27,28]. Such layers can act as a mediator between desired properties of the top coating layer and problematic adhesion mechanism on the interface.

## 2. Coating Deposition Techniques 

### 2.1. Physical Vapor Deposition (PVD)

PVD refers to a class of non-equilibrium techniques in which the compositional diversity of the formed films is not restricted by thermodynamics, allowing for the deposition of metastable solid solutions. PVD techniques are categorized into sputtering and evaporation based on the physical approach used to transfer the solid precursor material (the so-called target) into the vapor phase. PVD methods are line-of-sight processes, which implies that they only coat the region of the substrate that is immediately exposed to the target. The ability to employ temperature-sensitive substrates at low substrate temperatures near to room temperature is a significant benefit of PVD [29]. Reactive deposition methods are often used, in which a reactive gas (e.g., N_2_ or O_2_) is supplied into the deposition chamber in addition to the working gas (typically Ar) [30]. The substrate holder can be grounded, or a negative substrate bias voltage can be supplied, resulting in intense ion bombardment of the growing film, which enhances activation of film growth due to atomic scale heating. PVD films often have high defect densities, tiny grain sizes, and compressive stresses because of the extra kinetic stimulation of film formation [31]. Since they are so adaptable to changing market conditions, a wide variety of techniques and procedures, some of which are shown in Figure 1, have been developed and improved. Two of the most prevalent PVD processes for the deposition of thin films are sputtering and evaporation, as shown in Figure 2.

### 2.2. Chemical Vapor Deposition (CVD)

CVD refers to any process in which gaseous precursors are injected into a reaction chamber to create a coating [33]. The activation of dissociation and chemical surface reactions of the precursors is mostly accomplished thermally, which can be accomplished, for example, using a hot-wall reactor in which the substrates are heated indirectly. Hot-wall reactors are commonly used in the hard coating industry. Moreover, the substrates can be warmed up inductively or resistively in a cold-wall reactor or with laser aid. Another option is plasma-assisted CVD, which relies on kinetic activation [34]. In addition to the substrate temperature, which is in the range of 800 to 1150 °C in thermally activated CVD, the deposition pressure is a critical process feature [35,36]. Because the inner gas stream velocities, which determine the retention time and the thickness of the boundary layer, and thus the reactant diffusion to the substrate surface are pressure-dependent, the deposition pressure influences the homogeneity of the deposition process within the reactor. The intended reaction to generate a coating is a heterogeneous reaction on the surface. Although CVD is far more driven by thermodynamics than PVD, it is not an equilibrium process and also has kinetic limits, which define the deposition rate and have a substantial influence on the coating microstructure. There are two rate restrictions to consider: (I) surface response limitation and (II) mass transport limitation. At low temperatures or pressures, reaction happens slowly, the boundary layer is thin, diffusion is rapid, and reactants reach the surface quickly. When mass transport is the limiting factor, the diffusion rate of the reactants across the boundary layer is the decisive parameter. At high temperatures and pressures, mass movement is frequently limited, resulting in low gas velocity and thicker boundary layers. Deposition inside the surface reaction kinetics regulated regime is commonly preferred because it results in more uniform deposition within the reactor [34]. Homogeneous gas phase reactions often occur at temperatures higher than the decomposition temperatures of the species inside the reactor, resulting in powder formation, which is undesirable in a deposition process. The tremendous throwing power of CVD and hence the ability to coat complicated shapes is a significant benefit. Additional advantages include large batch sizes and high coating thicknesses. Aside from the high deposition temperatures, which limit the substrate materials available, another disadvantage of CVD coatings is that they frequently display tensile residual stress and a high surface roughness, both of which are undesirable in application and demand post-treatment [37].

### 2.3. Electrodeposition Coating

Material electrodeposition is a method of protection that involves the deposition of metallic ions on a substrate. A potential difference between the anode and cathode poles induces an ion transfer in the unit cell in this process. By accepting ions from the other electrode, a coating layer develops on the submerged sample after a while. Extensive research has been conducted on popular electrodeposition materials. The most often researched metals include but are not limited to Ni-P, Ni-P/Sn, Ni-P-W, Ag/Pd, Cu/Ag, Cu/Ni, Co/Ag, and Co/Pt [38,39]. These studies show that electrodeposited coatings greatly improve the corrosion characteristics of the substrate. Furthermore, this approach has shown promise for enhancing the adhesion of hard coatings [40]. In general, electrodeposition is divided into two processes: electrolytic deposition (ELD) and electrophoretic deposition (EPD), which are further explored in the sections that follow.

#### 2.3.1. Electrolytic Deposition (ELD) Coating

Electrolytic deposition (ELD) is an electrochemical method used on conductive surfaces to generate a dense metallic coating with a homogeneous thickness distribution. Substrate and deposition materials are selected as cathode and anode while placed inside an electrochemical unit cell. Metallic ions move toward the working electrolyte and then toward the substrate when a potential difference between the anode and cathode poles is applied. The deposition phase necessitates electrolyte supersaturation, which happens as a result of charging current in the circuit. During the coating process, the concentration of metallic ions in the electrolyte remains constant [41]. Although this technology is generally utilized for decorative and low-corrosion/wear applications, there have been reports of additional uses being developed, including optical, electronics, biomedical, high-temperature, and solid-oxide fuel cells [42]. Ceramic materials may be deposited on metallic substrates in a manner similar to the micro-arc oxidation (MAO) process by raising the potential difference in electrolytic unit cells. Tian et al. [43] deposited Ni-Co-Al_2_O_3_ on steel pipes and found that it significantly improved corrosion of the substrate exposed to oil sand slurry. Yang et al. [44] showed considerable corrosion and erosion increased corrosion resistance after depositing Ni-Co-SiC on carbon steel pipes subjected to oil sand slurry. Fayomi et al. [45] obtained similar findings on a Zn-Ni-Al_2_O_3_-coated mild steel substrate. Furthermore, Redondo et al. [46] used a dihydrogen phosphate solution to deposit a corrosion-resistant polypyrrole (PPy) layer on a copper substrate.

#### 2.3.2. Electrophoretic Deposition (EPD) Coating

Electrophoretic deposition (EPD) is another type of electrodeposition that produces thicker colloidal coating layers. Thin films produced on surfaces by coagulation of colloidal particles using an electric field in a unit cell similar to that used in ELD. EPD is a multi-phase approach that includes the following steps: (I) Electrophoresis is the process by which an external electric field pushes suspended particles in an electrolyte toward one electrode. (II) Moving particles congregate in one electrode and combine to produce a bigger coagulated particle. (III) The bigger particles settle on the electrode’s surface, which is a to-be-coated substrate. Eventually, a thick coating layer with a powder-shaped structure will be formed on the substrate. Densification methods (e.g., furnace curing, light curing, sintering, etc.) are advised to improve the protective layer’s quality. EPD has been used for a variety of applications, including coating, selective deposition, graded material deposition, porous structure deposition, and biological applications [47]. Borides, carbides, oxides, phosphates, and metals are widely utilized in EPD [48]. Castro et al. [49] used sol-gel and EPD to create corrosion-resistant coatings on stainless steel AISI 304 and observed two- and four-times improvements in corrosion resistance for each technique, respectively. Gebhart et al. [50] used chitosan to cover an AISI 316L stainless steel for biological purposes in another investigation. They found that this coating had a good influence on the corrosion behavior of the substrate. They also claimed that the applied electric field in EPD is the most important aspect in determining coating properties such hydrophobicity, thickness, and structure. Chen et al. [51] used graphene to cover TC4 Ti-alloy orthopedic implants. They discovered that graphene-coated artificial joint implants had a much longer life expectancy. They discovered that micro-cracks in coating surfaces were the cause of any corrosion on substrates. Fei et al. [52] investigated the wear resistance of EPD coatings and effectively deposited SiC particles on paper-based friction materials, achieving good wear enhancement.

### 2.4. Thermal Spray Coating

Thermal spray coating is a collective term of methods that use a plasma, electric, or chemical combustion heat source to melt a specific set of materials and spray the molten substance onto a surface to form a protective layer. These are dependable corrosion- and wear-resistant coatings. In this method, a heat source, which is often produced by chemical combustion or plasma discharge, warms the materials to a molten or semi-solid state before spraying them on the substrate with a high-speed jet. Thermal spray coating methods may produce thicknesses ranging from 20 m to several millimeters, which is much more than electroplating, CVD, or PVD procedures [53]. Furthermore, the materials that may be utilized as feedstock for thermal spray coatings range from refractory metals and metallic alloys to ceramics, polymers, and composites and have the ability to cover a reasonably large surface area of a substrate. Thermal spray coatings are classified according to their features and process criteria (Table 1). Plasma spray coating, high-velocity oxyfuel (HVOF), cold spray coating, warm spray coating, and arc wire spray coating are the most prevalent types [54,55].

## 3. A Comparison of Coatings on Cutting Tools

For cutting tool coatings, the most popular and latest options will be discussed below as well as the deposition process. In addition, novel tool geometries will be demonstrated, including those applied to the coated insert substrate to enhance chip removal and breakage. The cutting inserts on the cutting edges of modern cutting tools are commonly made of stainless steel. There are several coatings available, ranging from monolayers to nanocomposite coating materials, which generate a two-phase composition during the deposition process [75,76]. 

Hard coatings, or nitrides, carbides, borides, and oxides of transition metals, are commonly employed in tool coatings. TiN, TiAlN, CrN, ZrN, TiSiN, TiAlSiN, CrAlN, TiAlCrN, and cBN are some of the nitride coatings used on cutting tools. Carbide coatings include TiC, CrC, and WC. With its chemical inertness and excellent hardness, TiB_2_ is ideal for boride coatings. Because they have strong adhesion properties, they may be applied on tool steel [77,78,79]. The borocarbide coatings MoBC, WBC, and TaBC can also exhibit good fracture resistance and high hardness and extend the tool lifetime. In high-friction machining operations such as micro milling, boride coatings are used [10,80]. Al_2_O_3_ is a commonly used oxide coating. Diamond-like carbon (DLC), MoS_2_, and WC-C are also prevalent coatings for cutting tools. As a consequence of several lines of research that continue to compare coated tools to uncoated tools, it is known that coatings applied to milling inserts increase milling process efficiency. It all comes down to the material being machined and the desired results, such as a higher MRR or a higher-grade surface finish. To further understand the cutting capabilities of Nimonic alloy 75, uncoated and TiAlN-coated tungsten carbide micro end mills are compared. The authors determined cutting performance by considering tool wear, groove geometry, bulge formation, and surface quality under the similar wear conditions. The TiAlN-coated tools often outperformed the uncoated ones. As a result, the surface polish quality and slot geometry improved [81]. Masooth et al. [82] analyzed the surface roughness of an Al6061–T6 alloy end mill using uncoated and TiAlN coated carbide tools. Figure 3 shows the comparison of these two workpieces. The Taguchi technique was employed to optimize the process, and an analysis of variance (ANOVA) was utilized to establish that spindle speed was the most important factor in surface roughness. The employment of both uncoated and coated tools helped to improve the efficiency of the process. Results show that the coated carbide end mill produced a superior surface polish.

Martinho et al. [83] investigated the performance of comparable carbide insert coated tools with varying coatings. PVD (monolayer) AlTiN and CVD (multilayer) TiN/TiCN/Al_2_O_3_ coatings were employed. During the cutting process, the tool was monitored for vibration, wear, and the ultimate condition of the machined surface. Compared to CVD-coated inserts, PVD-coated inserts showed higher failures and were found to be brittle. Surprisingly, wear of the CVD synthesized coating did not change the machined surface. It was reported that PVD coatings exhibited a better adhesion to the substrate than the CVD coatings, which was conducted by the same researchers utilizing a GX2CrNiMoN266-7-4 super duplex stainless steel alloy to manufacture inserts and coatings [84]. 

There were no significant differences in surface quality between the PVD-deposited monolayer coating and the CVD multilayer coating; however, a severe wear and abrasion was observed in the former coating. Furthermore, it was found that the substrate was more severely damaged by wear once the coatings had worn away. Hosokawa et al. [85] evaluated the cutting properties of PVD-coated tools synthesized by filtered arc deposition (FAD). This deposition approach was used to create two novel kinds of films in this investigation. A high-speed milling procedure was used to apply these coatings on a pre-hardened stainless steel. In the study, TiCN and VN coatings were investigated. The TiCN coatings exhibited excellent hardness and adherence to the substrate. In the machining application, TiCN-coated tools were found to be more effective than coated tools available commercially, obtained by arc ion plating (AIP) or hollow cathode discharge (HCD) techniques. Tools deposited with VN coating exhibited an improved tribological and mechanical properties at high temperatures. Therefore, the VN-coated tools might be a proper choice when high temperature is inevitable. The effects of carbon and boron co-doping vs. doping alone on the microstructure and performance of as-deposited AlTiN, AlTiCN, AlTiBN, and AlTiBCN coatings was discussed by Mei et al. [86]. The results indicated that typical columnar crystal structures were observed with dopped coating. 

### 3.1. Diamond Coatings

In recent studies on coated cutting tools, diamond-like carbon (DLC) coatings have been a good solution for dealing with abrasion issues. These coatings are smooth with a typical coefficient of friction (COF), which often leads to a low wear rate. Therefore, DLC coatings can reduce the friction in the contact area and hence avoid early heat generation during the machining operation. DLC can also be combined with tough layers to increase the toughness of the coating. For instance, the characteristics of a three-layer CrN/CrCN/DLC have been evaluated [87]. Furthermore, these coatings were compared to a TiAlSiN coating and an untreated surface. It was 58 times more resistant to wear than an untreated surface. Ucun et al. [88] investigated tool performance during machining in terms of tool wear, surface roughness, burr formation, and cutting forces, as illustrated in Figure 4 and Figure 5; because of the decreased friction of the DLC coating, the cutting forces for the coated tool test were lower than those for the uncoated tool. When milling with an untreated tool, burr creation increases as the cut length grows. At low cutting lengths, there was no discernible difference in burr development between uncoated and coated tools. To conclude, the DLC-coated tools give rise to a reduced COF and wear, hence providing a smoother polished surface, as shown in Figure 6. 

It has recently been reported that CVD diamond cutting tools have been developed for use in the micro milling of oxygen-free copper. Because of their high hardness, diamond coatings are a viable micro-milling tool. Using laser-induced graphitization and precision grinding, Zhao et al. [89] suggested a unique approach for the fabrication of these micro-cutting tools. These micro-cutting tools claimed to show better performance than commercially available coated cemented carbide micro-cutting tools used for oxygen-free copper machining. Additionally, the CVD diamond tools exhibit a superior surface polish and reduced cutting forces throughout the milling operation. 

Several studies have recently documented the milling process and diamond coatings. Wang et al. [90] examined the cutting performance of several high-speed machining of graphite molds using diamond-coated tools. The microcrystalline diamond (MCD), sub microcrystalline diamond (SMCD), nanocrystalline diamond (NCD), and micro/nano-crystalline composite diamond (MCD/NCD) coatings were tested to assess their ability against scratching. While NCD was found to have the lowest friction coefficient, MCD exhibited a better adhesion to the substrate, as shown in Figure 7. Furthermore, as compared to MCD and NCD, the SMCD coating’s performance was subpar utilizing these parameters. However, the MCD/NCD composite showed strong adhesion to WC-Co. The friction was decreased because the top layer of the coating was NCD, which has a lower surface roughness and finer diamond particles. 

The use of MCD/NCD composite coated tools in the milling of hot bending graphite molds and related materials establish a compelling case for their use. Wang et al. [91] reported the same occurrence in their investigation. This coating has a reduced friction coefficient when compared to the monolayer NCD coating owing to a lack of adhesion to the substrate. It was found, however, that the multilayer diamond film with surface coating of NCD layer (MNMN-CD)-coated tools had 7.5 times longer lifetime than the monolayer-coated tools.

### 3.2. Carbon Nano Tubes (CNTs)-Coated Tool 

Currently, carbon nanotubes (CNTs) have attracted significant attention in PVD hard coatings due to their exceptional tensile and shear strength. Additionally, a faster heat dissipation during machining is aided by their high thermal conductivity (4000–6000 W/mK) [92]. The wear characteristics and machinability features of CNT-coated inserts were studied by Pazhanivel et al. [93]. These inserts exhibited CoF, which can point out higher machinability and better surface polish.

Hirata et al. [94] studied the sliding friction characteristics of CNTs deposited on silicon, cemented carbide, and silicon nitride substrates, and the lubrication and adhesion properties of CNT-coated substrates were shown to be superior on substrates with surface porosity. Borkar et al. [95] found that pulsed electrode deposition of CNT coatings resulted in a considerable increase in wear resistance compared to pure nickel coatings. 

At various coating conditions, Chandru et al. [96] deposited carbon nanotubes (CNT) over the HSS tool and evaluated the machining performance of the coated tool with respect to the important machinability aspects such as cutting tip temperature, cutting forces, surface roughness, and tool wear and life. Due to the excellent mechanical and thermal properties of CNTs, drastic reductions in cutting tip temperature and cutting forces of 70 to 80% were observed for coated tools. However, there was no significant effect of coating on surface roughness at lower levels of cutting conditions, but significant improvement was observed at higher levels of cutting conditions. The tool wear and life analysis explored the steady improvement to a remarkable extent. Very specifically, under elevated machining conditions, tool life investigations concluded the fact that the CNT-coated tool has a tool life of more than 60 min with negligible failure.

The study achieved by Chenrayan et al. [97] investigated the deposition of carbon nanotubes (CNT) over the HSS tool through PECVD. It also evaluated the coated tool’s machining capability in terms of mechanical performance, surface quality, cutting parameters, temperature, cutting forces, friction coefficient, and service life. The analysis of experimental outcomes for three different cutting environments shows that the CNT-coated tool represent a viable candidate for machining of harder materials. Furthermore, dramatic reduction of the cutting tool tip temperature and cutting forces due to the excellent mechanical and thermal properties of CNTs was recorded. This demonstrates that the CNT-coated tools outperform the DLC-coated tools in terms of applicability, as shown in Figure 8 and Figure 9.

The most critical finding of other studies can be summarized as follows: CNTs deposited using the PECVD technique have exceptional adhesive strength. The scratch test findings show that the adhesive stress value is greater than 75% of the yield strength of the substrate material. CNT coating reduced heat generation by 70–80% compared to the uncoated HSS tool and improved the metal cutting performance at elevated temperatures. In diverse cutting circumstances, coated tools showed a 70–80% reduction in cutting forces. In all machining settings, the tangential cutting force is larger than the feed and radial forces. As a result, the observations validate the theoretical reality that cutting parameters increase cutting forces. However, the coating had a minor effect on surface roughness at the lower cutting level. 

Coated tools can be used to manufacture challenging materials at higher cutting rates since they minimize surface roughness by 30–40%. The amplitude of surface texture peaks and valleys rises with cutting conditions for both coated and uncoated tools. According to rigorous AFM and optical microscope measurements, the CNT-coated tool has a tool lifetime of >60 min even in the severe cutting conditions. In all cutting circumstances, the wear of coated tool flank was decreased by 60–70%. It was also observed that the rate of flank wear increased gradually for coated tools in the same conditions. Uncoated tools failed catastrophically in poor, moderate, and extreme cutting conditions at 24, 18, and 7 min, respectively, when the coated tool survived even for >60 min. However, the homogeneity in the CNT dispersion of the CNT-coated tool resulted in a dramatically shortened tool life [98].

### 3.3. AlTiN and AlTiSiN Coatings

A series of full studies on the surface morphology, hardness, surface section morphology, and wear resistance of AlTiN/AlTiSiN coatings were carried out [99,100], and the effects of different modulation periods on the microstructure, mechanical properties, and tribological properties of the coatings were systematic in order to improve the mechanical and tribological properties and expand the microstructure and properties of AlTiN/AlTiSiN coatings by arc ion plating. The influence of the modulation period on the microstructure and characteristics of TiN/TiAlN multilayer coatings was studied by Yongqiang et al. [101]. In all TiN/TiAlN multilayer coatings, XRD examination revealed a significantly preferred orientation (111) plane. It explains why this result occurred, which might be due to an increase in the inner tension as the coating grows. The rapid atom movement was aided by the powerful ion bombardment, which allowed them to easily align along the densely packed (111) direction. The nano hardness of TiN/TiAlN multilayer coatings, on the other hand, achieved a maximum of 38.9 GPa at a modulation period of 164 nm. The TiN/TiAlN multilayered coatings had greater hardness values, between 28.9 and 38.9 GPa, than the TiN and TiAlN monolithic coatings. The nanoscale multilayer design is responsible for this improvement. Because the individual layers of TiN and TiAlN had identical shear moduli, the coherent interface generated by epitaxial growth of TiAlN and TiN had an alternating strain field in the multilayers due to lattice misfit, which may prevent dislocation motion and promote multilayer strengthening. According to this study, the TiN/TiAlN multilayer coatings had the best adhesion strength at a modulation period of 54 nm. Tuffy et al. [102] investigated the influence of TiN coating thickness applied by the PVD method on tungsten carbide insert machining performance in dry hard turning of AISI 1040 steel. The TiN layer with a thickness of 3.5 mm had the best cutting efficiency, with a tool life of roughly 40 times. Sargade et al. [92] investigated the effect of TiN coating thickness on cemented carbide tools deposited between 1.8 and 6.7 mm during dry turning of hardened C40 steel. The critical load for scratch testing grew until the thickness reached 4 mm and then decreased. They discovered that a 4 mm thick covering increased the tool life of untreated cemented carbide by 9.5 times. The characteristics and structure of TiAlN/CrAlN- and TiAlN-coated cermets were investigated by Yang et al. [103]; during the machining of 9CrSi_2_Mn steel, SEM, EDS, and scratch tests were used to investigate cutting performance, wear mechanism, and adhesive strength. The TiAlN coating has a hybrid columnar structure that resists adhesive failure better than the TiAlN/CrAlN coating. The flank wear was more severe in TiAlN/CrAlN coating than in TiAlN coating. Kumar et al. [104] investigated the effect of different AlTiN (2–4 mm) thicknesses on Al_2_O_3_-TiCN mixed ceramic inserts coated by cathodic arc evaporation while hard twisting AISI 52,100 steel (62 HRC). The application of coating resulted in a considerable increase in machinability, with a coating thickness of 4 mm exhibiting higher machining performance.

Colombo-Pulgarin et al. [105] used laser powder bed fusion (LPBF) to manufacture Inconel 718 (IN718) specimens, which were then PVD-coated with TiN and AlTiSiN to physically and chemically characterize their surface. Microhardness tests and chemical analysis using glow discharge optical emission spectroscopy (GDOES) were carried out in this regard. The mechanical performance of both coatings as well as their surface and morphological properties were assessed using roughness and wear behavior studies. The experimental findings allowed for a comparison of the two coatings tested with the LPBF-processed substrate (Table 2). The microhardness of AlTiSiN and TiN was found to be greater than that of the substrate materials. Between the two, the former has a greater Vickers microhardness than the latter. In a wear process, AlTiSiN outperformed TiN in terms of wear resistance. When the identical test-time values of both coatings were compared with the rubber wheel, it was discovered that AlTiSiN had a lower relative weight loss when the weight loss of the substrate material was taken into account. The greater the mechanical adherence of the coating to the substrate material, the lesser the comparable weight loss.

### 3.4. Boron Nitride Coatings

In mechanical cutting and grinding, cubic boron nitride (cBN) is a synthetic wear-resistant substance with a hardness comparable to diamond [139]. Furthermore, cubic boron nitride particles have superior thermal stability to diamond. To further increase the wear resistance of the alloy, laser cladding (LC) technology was utilized to create (cBN)/Ti6Al4V and Ni-plated (cBN)/Ti6Al4V composite coatings on Ti6Al4V substrates. Cubic boron nitride particles lose their raw characteristics when exposed to laser light in a molten pool. Composite coatings’ wear resistance, worn track shape, and microstructures were studied with scanning electron microscopy (SEM), as was the interface bonding between cubic boron nitride nanoparticles and the Ti6Al4V matrix (WTM-2E). By comparing the two coatings, it was discovered that the Ni-plated cubic boron nitride/Ti6Al4V composite coating had many fewer thermal flaws. The composite coating’s cubic boron nitride particles were spared heat break thanks to a Ni plating applied to their surfaces. A TiN reaction layer is produced between cubic boron and the Ti6Al4V matrix, thus preventing further disintegration of cubic boron nitride particles. Between the cubic boron nitride particles and Ti6Al4V, a TiN reaction layer formed. The composite coating’s wear resistance was improved by nickel plating the cubic boron nitride particles. 

Due to a lower cutting zone temperature, a protective coating’s physical and mechanical characteristics are evaluated to see how this affects cutting tool life in turning hardened steel [140]. In the amorphous state, boron nitride is believed to be a solid lubricant in the tool-cutting chip contact that reduces the contact length and cutting force, resulting in a gain in tool life and dependability, particularly during the run-in period. According to physico-mechanical and thermal studies, the hardness and Young’s modulus, coefficient of thermal conductivity, thermal capacity, and friction of the coating on the ShKh15 steel were found to be 15 GPa, 200–220 GPa, 70 W/mK, 800 J/kg K and 0.3, correspondingly. When cutting rate and feed are increased while using a tool with a protective coating on the working surfaces, the total contact length decreases as the friction coefficient decreases. It is feasible to extend the tool life and dependability, especially during the run-in stage, by applying a coating of amorphous boron nitride to the area where a chip meets the cutting pressures. Tool wear is reduced by 22–25% with the boron nitride coating while turning hardened steels due to a difference in the thermobaric loading compared to tools without the coating. Boron nitride nanotube (BNNT) has emerged as an outstanding mechanical, chemical, and thermal material for ceramic and metal matrix reinforcement. In this study, the atmospheric plasma spraying approach was used to add 5% BNNT to the alumina (Al_2_O_3_) matrix in this work [141]. 

BNNTs have been able to withstand the plasma plume’s tremendous temperatures. It has been discovered that BNNTs can increase the relative density of an Al_2_O_3_ coating from 90% to 94% while also increasing the hardness, elastic modulus, and fracture toughness of the BNNT-reinforced coatings by 117%, 35%, and 51%, respectively. With the addition of BNNT, the critical energy release rate rose from 3.90 1.18 J/m^2^ to 6.50 1.22 J/m^2^. The increased attributes are ascribed to improvements in density, homogeneous distribution of BNNTs, and toughening mechanisms such as BNNT truss, nanotube pull-out, and fracture bridging. This BNNT-enhanced Al_2_O_3_ coating might be a suitable material for automobiles, aircraft, high-speed cutting tools, and bioimplant applications. Sugihara et al. [142] investigated high-speed milling of Inconel 718 with CBN cutting tools to understand the tool wear behavior better and extend tool life. The following results were gathered: Inconel 718 is a difficult material for machining at high speeds because of its high melting point, which causes a diffusion wear phase at the beginning of cutting and a following cracking step due to the creation of a thick adhesion layer on the worn flank face and its flaking, which accelerates CBN tool retreat. The anchoring effect of the textured surface was used to design new CBN cutting tools with textured flank faces, utilizing a femtosecond laser in order to stabilize the adhesion layer on the flank face and prevent it from flaking. A series of cutting experiments revealed that the CBN-ORE cutting tool significantly reduced tool retreat compared to a conventional CBN cutting tool without surface texture, indicating that surface roughening on a flank face is a promising approach for enhancing CBN durability during high-speed milling of Inconel 718, as demonstrated by a series of cutting experiments shown in Figure 10 and Figure 11.

Azinee et al. [143] coated the instrument with cubic boron utilizing electroless nickel co-deposition. The Ni-CBN surface coating was applied electrolessly using nickel, using chemical methods to integrate the ceramic CBN particles. Electroless nickel can minimize coating costs and time. Corrosion-, wear-, and abrasion-resistant coatings have been widely employed in the industry. This study compared the surface tolerance of Ni-CBN HSS tool substrates to uncoated HSS tools for cutting aluminum alloy 7075. Using Taguchi L9 experiments, the authors chose to examine cutting speed, feed rate, and depth of cut. The lowest and most significant flank wear measures differ significantly for coated surfaces, as shown in Figure 12. Uncoated surfaces are 1.154 µm at the roughest and 0.42 µm at the smoothest. The maximum and minimum Ra for coated substrates is 0.787 µm and 0.251 µm. At the end of the test, the Ni-CBN HSS cutting tools had the longest tool life of 195 min compared to the uncoated 143 min. Thus, Ni-CBN HSS tool end mills can reduce tool wear in general.

### 3.5. Tungsten Carbide-Cobalt (WC/Co) Coatings

It is a superb anti-friction and wear substance used in the preparation of hard coatings and in industrial applications. The University of Connecticut used high-temperature flame spraying to generate nanostructured WC/10Co coatings with excellent hardness and bonding strength in 1994 [144]. Since then, nanostructured WC/Co coatings have become a significant issue of discussion. For the most part, traditional WC/Co coatings are only melted on the surface, while nano-WC/Co coatings are melted in the complete volume. Nano-WC cermet powder has a denser coating, granular strengthening, improved toughness and hardness, better adhesion, more strength, greater strength and crushing resistance, and improved wear resistance. Together with sophisticated surface modification technology and 3D intelligent programming technology, they shield the cutting tools from abrasion and extend the tool’s service life [145]. The WC phase in the coating is more equally distributed in the Co and Cr phases at a single high temperature. There was a considerable improvement in the microhardness value as well as a more uniform distribution of microhardness. 

The coating’s wear and sediment erosion resistance have been substantially enhanced. Myalska et al. [146] used mechanical blending and ultrasonic-aided mixing to combine WC/17Co raw material powder with various quantities of TiC nanoparticles and thermally spray them onto a carbon steel surface using the high-velocity air-fuel (HVAF) process. In the Co matrix, because the lower temperature of the HVAF spraying process slowly reduces powder oxidation during spraying, and consequently, phase composition and microstructure of powder are preserved, TiC was slowly oxidized but did not breakdown or dissolve. Nanoscale TiC also slowed the dissolution and decarburization of WC, making it more resistant to corrosion. When the coating was cured and slid at 400 °C, clusters of TiC nanoparticles protruded from the coated surface and were responsible for the majority of the contact stresses. Because of this, the wear rate decreased. Friction oxidation debris clusters (based on CoWO_4_ and WO_3_-2H_2_O) developed on the worn track and assisted in lowering the friction coefficient by eliminating direct contact between the WC/Co surface and the matching body. The structure, mechanical characteristics, and wear parameters of nano-WC/Co coatings generated by the high-velocity oxygen-fuel (HVOF) technique were examined at high temperatures. Studies showed that the nano-WC/Co coating had a dense structure, and the friction coefficient fell progressively with increasing temperature. The rate of wear dropped first but subsequently began to rise. 450 °C has the lowest wear rate. The friction coefficient and wear rate might be reduced by a continuous oxide coating of WO_3_/WO_3_-CoWO_4_ on the worn surface. Utilizing detonation spray coating (DSC) on mild steel (MS) substrates, Babu et al. [147] investigated the microstructure and phase composition of nano-WC/12Co raw materials and coatings. 

Using SiC as a grinding media, an abrasive wear test was conducted, and the coating wear rate was estimated. Hardness and Weibull modulus was discovered to be altered by variations in indentation load and coating microstructure. The number of notches necessary to obtain the specified hardness value was established based on the results, and the structure–performance connection was investigated. To obtain the best performance and performance, Weibull parameter-based strategies were proposed.

### 3.6. Hafnium and Vanadium Nitride Coatings

Hafnium and vanadium nitride multilayer coatings [HfN/VN]_n_ coatings were investigated as a replacement for TiN coatings for gear cutting tools in the 1980s due to their higher thermal barrier and hot hardness, but their production costs and times were so high due to TiN deposition processes at the time that new PVD coatings on a base of transition metal nitride compounds were required [148]. HfN monolayers, VN monolayers, and [HfN/VN]_n_ coatings were deposited onto a variety of substrates using cutting-edge equipment, and their microstructure, chemical composition, morphological topography, and layer thickness were studied to identify properties such as high hardness, high modulus of elasticity, and a critical load increase; these properties result in outstanding mechanical behavior. 

According to Duran et al. [149], deposition of the HfN monolayer coating enhances the surface smoothness of the machined component, exhibits homogeneity of the manufactured piece, and decreases surface microcracks, thus preventing probable early failures in the manufacturing process [60]. An HfN monolayer-coated cutting tool has several advantages at the industrial level, including boosting the quality of made products, lowering production time and costs, as well as minimizing the adhesive wear caused in the tribological pair [150]. Using a computer numerically controlled (CNC) machine, the temperature of the steel bar and the tool was monitored; the results showed that the Monolayer HfN coating reduces tool wear and improves the surface smoothness of the machined item, which was reflected in the process temperature change, adding HfN coating to a tool can extend the tool’s life, improve quality, and save manufacturing costs. 

The use of [HfN/VN]_n_ multilayer coatings on the flank and rake faces of HSS cutting tools improved tribological compatibility, reduced friction coefficient, and increased tool life, according to Navarro-Devia et al. [151]. Using a cutting tool with [HfN/VN]_n_ multilayer coatings enhance workpiece surface integrity by reducing friction and increasing thermal stability proportional to bilayer number. Because the coating affects the cutting process, it improves corrosion resistance. Using [HfN/VN]_n_ multilayer thin films as protective coatings on an HSS cutting tool can increase tribological compatibility in low carbon steel turning. The coatings’ outstanding tribological properties not only increase cutting tool wear resistance but also reduce cutting forces by reducing friction and acting as a thermal barrier. A thinner chip reduces the chip compression ratio by decreasing stresses and temperatures at the interface, and softer surfaces in milled steel improve workpiece quality by increasing resistance to corrosion. With a bilayer number (n), [HfN/VN]_n_ multilayer coatings promote tribo-pair interactions, enhancing workpiece surface integrity and tool life. 

Escobar et al. [152] deposited HfN and VN films using magnetron sputtering. The elastic modulus of HfN (224 GPa) is larger than that of VN (205 GPa), which is evident in variations in plastic deformation resistance values. The tribological pair (steel pin and nitride film) had low friction coefficients for both films, 0.44 for HfN and 0.62 for VN, and critical loads for HfN and VN films were about 41 and 34 N, respectively. SEM identified the wear processes in both films: adhesive wear in HfN and abrasive wear in VN. Finally, the HfN films had 9% better mechanical characteristics than the VN films and 18% better tribological properties (friction coefficients) than the VN materials. Escobar et al. [153] investigated the wear and tribological behavior in great detail. As bilayer periods in the coatings were reduced, an increase in hardness and elastic modulus of up to 37 GPa and 351 GPa was observed. In terms of friction coefficient (0.15) and critical load (72 N), the sample with a bilayer period of 15 nm and a bilayer number of n = 80 had the lowest friction coefficient and the greatest critical load, respectively. WC inserts were utilized as substrates to enhance the mechanical and tribological characteristics of [HfN/VN]_n_ coatings as a function of increasing contact numbers and to achieve improved efficiency in various industrial applications, including machining and extrusion. 

Cutting experiments with AISI 1020 steel (workpiece) and bilayer number and bilayer period were carried out to evaluate wear as a function of the bilayer period and number of bilayers. Compared to uncoated tungsten carbide inserts, WC inserts coated with [HfN/VN]_80_ showed a decrease in flank wear of around 24%. New coatings for tool machining with high industrial performance may be achieved by employing [HfN/VN] multilayers.

With bilayer periods ranging from 1200 to 15 nm, Escobar et al. [154] succeeded in depositing multilayered hafnium/vanadium nitride systems onto AISI 4140 steel substrates using the multi-target magnetron sputtering approach. Images from the SEM (Figure 13) and TEM (Figure 14) revealed layers of evidence with a well-defined and consistent periodic pattern. Over uncoated AISI 4140 industrial steel, multilayered coatings significantly improved corrosion characteristics, strengthened the material’s defenses against rust, and lowered the corrosion rate. Compared to untreated steel, the multilayered coatings demonstrated improved resistance to corrosion due to an interface effect that created a protective layer over the steel substrate. Because of the decreased porosity and shorter bilayer period (15 nm) of the [HfN/VN]_80_ multilayered coatings, these reactive species were unable to diffuse as easily into the substrate. Therefore, when the bilayer period of the multilayer coating was reduced, the best conductivity was discovered.

In the study developed by Caicedo et al. [155], they used carbon nitride coatings (V-C-N) to study the mechanical and tribological properties of V-C-N-coated industrial steel (AISI 8620). Using a magnetron sputtering method, the coatings were applied on silicon (100) and steel substrates by changing the applied bias voltage. Nanoindentation, pin-on-disk, and scratch test curves were used to evaluate the hardness, friction coefficient, and critical load of V-C-N surface material. As a function of bias voltage deposition, mechanical and tribological behavior in the VCN/steel system showed an increase in hardness of 58% and a reduction in friction coefficient of 39%. This shows that VCN coatings might be a potential material for industrial applications such as high-speed cutting (HSC) covered with a VCN layer evaluated in machining operations against heat-treated steels (AISI 8620).

Staia et al. [156] studied the tribological behavior of milling tools covered with HfN. The early wear behavior of multilayer HfN coatings is documented in laboratory friction and wear experiments against WC as a tribological pair. Vickers micro indentation experiments were also performed to characterize the coatings’ mechanical characteristics, evaluating the coating-substrate composite hardness. A model recently constructed by one of the authors provided information on the absolute hardness of HfN and TiCN monolayer films. These data are used to determine the HfN/TiCN multilayer coating characteristics. For the same number of cycles, the wear volume of single-layer HfN coatings is roughly 15 times that of HfN/TiCN coatings. 

### 3.7. ZrO_2_ Nanostructured Coatings

Nano-zirconia coatings are often used as thermal barrier coatings because to their low temperature coefficient, high thermal expansion coefficient, and strong stability at high temperatures. The tie layer and zirconia coating make up the thermal barrier coating. Wang et al. [157] created nanostructured ZrO_2_-8 wt. % Y_2_O_3_ (8YSZ) thermal barrier coatings (TBCs) on steel substrates using atmospheric plasma spraying. When it comes to elastic modulus, nanostructured 8YSZ coatings were shown to have a lower value than conventional 8YSZ coatings, but their elastic recovery rate was greater. The uneven and chaotic distribution of fractures in typical TBC may be clearly seen in Figure 15. Cracks appear to be very tiny and rare, with no clear development direction in nanostructured TBC. This may be due to the increased fine grain and durability of nanostructured coatings. Yin et al. [158] employed FeAl powder, ZrO_2_ nanoparticles, and CeO_2_ additives to spray dry innovative multi-component raw materials. Plasma spraying was utilized to cover 1Cr18Ni9Ti stainless steel with FeAl/CeO_2_/ZrO_2_ nanocomposite coatings and1Cr18Ni9Ti stainless steel treated with FeAl/CeO_2_/ZrO_2_ nanocomposite coatings through plasma spraying. Both a Vickers micro indentation tester and a ball disc sliding wear friction mill were employed. In this study, the mechanical, friction, and wear characteristics of nano-composite coatings and pure FeAl coatings were studied and assessed. The microstructures and mechanical characteristics of the two coatings were examined in relation to their wear mechanisms. The strengthening effect of ZrO_2_ nanoparticles may explain the increased hardness, fracture toughness, and wear resistance of the nanocomposite coatings compared to pure FeAl coatings. 

Using a plasma spray ZrO_2_/B_2_O_3_/Al system, Zhang et al. [159] examined the impact of ZrO_2_ particle size on the structure and characteristics of ZrB_2_-containing composite coatings. Plasma spraying resulted in the formation of ZrB_2_ from the reaction of ZrO_2_, B_2_O_3_, and Al. When plasma spraying with nano ZrO_2_ particles, more ZrB_2_ was generated than when using micro ZrO_2_ particles, indicating that the reaction degree between the ZrO_2_/B_2_O_3_/Al composite powder and the nano-ZrO_2_ particles was greater. There was a layered structure to the plasma-sprayed ZrO_2_/B_2_O_3_/Al composite powder and nano-ZrO_2_ coatings. There was an excellent distribution of ZrB_2_ particles, and the microstructure was dense. Compared to micro-ZrO_2_, nano-ZrO_2_ coatings have substantially greater hardness and toughness.

### 3.8. Al_2_O_3_ and Al_2_O_3_/TiO_2_ Nanostructured Coatings

There are several military and industrial uses for nanostructured Al_2_O_3_ and its composite coatings. Several countries’ naval fleets are now using Al_2_O_3_/TiO_2_ nanocoatings as an anti-friction and wear material. Using plasma sprayed multi-layer coatings, Daroonparvar et al. [160] were able to effectively coat magnesium alloys. In the experiment, coatings of NiCrAlY, nano-Al_2_O_3_-13% TiO_2_, and nano-TiO_2_ were applied to the magnesium alloy matrix in the experiment and generated three plasma spraying coatings. An investigation of the corrosion behavior of these three distinct plasmas spraying coatings was conducted. According to the findings of the study, the novel coating (NiCrAlY/nano-Al_2_O_3_/13% TiO_2_/nano-TiO_2_) reduced corrosion and increased the impedance value of magnesium alloy, according to the novel coating. Certain micro-holes in the coating may be filled by a layer of titanium dioxide, which slows down the corrosion rate of Al_2_O_3_-13% TiO_2_. There are three layers of NiCrAlY/nano-Al_2_O_3_-13% Ti/nano-TiO_2_ coating. Spray drying, heat treatment, and plasma treatment were all effective methods for producing nanocomposite powders, according to Yang et al. [161]. The microstructure and characteristics of Al_2_O_3_/TiO_2_ nanocomposite powders were examined because of processing technique and nano additions. The Al_2_O_3_/TiO_2_ nanostructured composite powder was made by spray drying and heat treatment, and it had nanoparticle sizes and was very spherical. An amorphous inter-crystalline network layer rich in Ti, Zr, and Ce encircled the alfa-Al_2_O_3_ colony in the plasma-treated nanocomposite powder, resulting in a three-dimensional network structure. Plasma treatment resulted in spherical powders with a smooth surface and dense microstructures because of their quick melting and solidification. It was thus possible to increase the fluidity and density of plasma-treated powders.

### 3.9. Cr Nanostructured Coatings

Wear-resistant coatings based on the (Ti,Cr,Al)N system is commonly utilized to enhance the performance qualities of metal cutting tools. Incorporating silicate (Si) with the aforementioned coating composition further enhances its performance [162,163]. (Ti,Cr,Al,Si)N coatings with up to 60% Al and Si content were found to have a cubic structure of the NaCl type. The hexagonal (wurtzite) structure dominates when the content is high. Coatings with a maximum (Al,Si) concentration of 56% were found to have a maximum hardness of 33 GPA, which concurrently retains the cubic structure [164,165]. According to the research of Yamamoto et al. [166], (Ti,Al,Cr)N coatings can have both hexagonal and cubic phases. This coating’s oxidation resistance reached 1000 °C (as opposed to 850 °C for the (Ti,Al)N coating). When heated to 1000 °C, the (Ti,Al,Cr)N coating’s dominating crystal structure altered from cubic to hexagonal. There was a drop in lattice parameter in this situation because of the probable migration of Al from cubic to hexagonal grains. The coating’s microhardness decreased as a result. At room temperature, the hardness of coatings containing varying amounts of different elements was almost identical (about 29 GPa). When the temperature was raised to 1000 °C, the (Ti_0.27_Cr_0.11_Al_0.62_) N coating had the highest hardness (27 GPa). The hardness of the material was increased by 31 GPa by heating it to 800 °C. Grain sizes were decreased to 5–10 nm with the addition of Al to the coating mixture.

Introducing Si to the (Ti,Cr,Al)N coating’s composition (2–3%) might alter the coating’s characteristics, as Yamamoto et al. [167] discovered. Researchers found that adding Si to the coating formulation enhanced its hardness and resistance to thermal oxidation for temperatures over 1000 °C. As Al,Si surpassed 60%, the hexagonal phase content rose, and the cubic phase content dropped. During research of temperature oxidation of this layer, coatings with high Si content were found to have formed a thin oxide layer between the basic oxide and the coating. According to the scientists, the barrier layer played an important role in preventing oxygen and metal ions from entering the coating structure. The composition of the (Ti,Al,Cr)N coating had no influence on its tribological factors at room temperature, but at 650 °C, the coating with a high concentration exhibited excellent tribological properties leading to the incorporation of a chromium oxide film, which is preferable in anti-friction features to films of aluminum and titanium oxides. Using a coating with a high Cr content resulted in a lower tool life at low cutting speeds (and, accordingly, at relatively low temperatures). Nonetheless, the coating with a Cr concentration of roughly 50% demonstrated greater wear resistance when cutting speed exceeded 100 m/min [168]. Ti_0.22_Cr_0.22_Al_0.44_Si_0.12_N’s cubic structure was maintained until the temperature reached 1000 °C, while Ti_0.20_Cr_0.20_Al_0.55_Si_0.05_N’s cubic and hexagonal phases were already disintegrated at 900 °C in thermal stability investigations. Si atoms were included in the architecture of the phases to slow down their disintegration. The hardness of the Ti_0.20_Cr_0.20_Al_0.55_Si_0.05_N composition rose from 32 to 34 GPa when the temperature was raised to 1000 °C, due to the coherence deformation between the matrix and the newly created cubic domains. The hardness of the Ti_0.22_Cr_0.22_Al_0.44_Si_0.12_N composition decreased from 34 GPa to 32 GPa when the temperature was raised to 1000 °C [162]. When Ti and Al were added to the CrN structure, the B1–NaCl-type cubic structure with the desirable orientation was formed, according to Tam et al. [169]. In the (Cr_0.40_, Ti_0.1_, Al_0.1_)N coating, the CrN, TiN, and AlN nitride phases were found, but no Cr2N phase was found. The (Ti,Al, and Cr)N coating was also shown to have higher wear resistance than the CrN, (Cr,Ti)N, and (Cr,Al)N coatings. At room temperature, the friction coefficient of the chosen coating was between 0.4 and 0.5. A comparison of the characteristics of the CrN, (Cr,Ti), (Cr,Al), and (CrTi,Al)N coatings was carried out by Wang et al. [170]. In the investigation, it was observed that the (Cr,Ti,Al)N coating had smaller grains and a much greater hardness and fracture resistance than other coatings. It was found that the adhesion strength of the provided coating to the carbide substrate was much decreased, and considerable residual compressive stresses were also found, resulting in the production of annular fissures. The characteristics of (Ti,Al) N–CrN-type nanolayer coatings were also examined. 

Al_x_Ti_1_−_x_N and CrN have been widely used as a protective coating material in many types of tools and mechanical components because of high wear performance and high temperature resistance. Chang et al. [171] investigated the monolayer of Al_0.63_Ti_0.37_N and the nanolayer of Al_0.63_Ti_0.37_N/CrN. Thermal stability and resistance to oxidation at temperatures between 700 °C and 1000 °C were examined. An oxide film was generated on the Al_0.63_Ti_0.37_N and CrN layers that was significantly thinner than the one on the Al_0.63_Ti_0.37_N coating, even at temperatures of up to 1000 °C, whereas the Al_0.63_Ti_0.37_N coating had a thicker oxide film. The nanolayer (Al,Ti,Si)N and (Cr,Si)N coating was explored by Yang et al. [172]. The B1–NaCl type structure was also present in this coating. With a rise in silicate percentage, coating hardness (up to 35 GPa) increased, but with an increase in silicate content of over 8%, coating hardness and adhesive binding strength decreased. 

Fukumoto et al. [173] investigated the nanolayer (Ti,Cr,Al)N/(Al,Si)N coating. When it came to layer structure, it was discovered that the (Ti,Cr,Al)N layer had a hexagonal structure, whereas the (Al,Si)N layer was cubic. When the nanolayer thickness was reduced, the coating’s hardness rose. Furthermore, it was discovered that the coating was able to withstand temperatures of up to 1100 °C. Chang et al. [174] investigated nanocrystalline CrAlSiN and CrTiAlSiN coatings, which were deposited by using a cathodic-arc deposition system with lateral rotating arc cathodes. Titanium, chromium, and Al_89_Si_11_ cathodes also were used for the deposition of CrAlSiN and CrTiAlSiN coatings. Cr_0.36_Al_0.57_Si_0.07_N and Cr_0.40_Ti_0.22_Al_0.36_Si_0.02_N nanocrystal coatings samples were evaluated. Grain diameters of 8–10 nm was found in these coatings. TiO_2_, Al_2_O_3_, and Cr_2_O_3_ are produced on the surface of the Cr_0.40_Ti_0.22_Al_0.36_Si_0.02_N coating when it is heated to 900 °C. Al_2_O_3_ and Cr_2_O_3_ were determined to be the principal components of a protective oxide layer that prevented oxygen from entering the coating. The results indicated that Cr_0.36_Al_0.57_Si_0.07_N with higher Al, and Si contents possessed superior oxidation resistance than Cr_0.40_Ti_0.22_Al_0.36_Si_0.02_N. Moreover, the oxidation resistance increased as the (Al, Si) N level in the solution rose.

Zhang et al. [175] looked at the nanolayer Cr,Al,Si,N coating as a possible alternative. Cr,Al,Si hexagonal and cubic nanolayers were alternated in the coating’s structure, with a nanolayer thickness of roughly 7 nanometers. The provided coating was discovered to have an incredibly high hardness rating (up to 52 GPa). The provided coating, on the other hand, demonstrated great plastic deformation resistance. In the ball-on-disk test, the coating had an extremely low friction coefficient of 0.1–0.2. These numbers were explained by the high cohesive energy of interfacial bonds and an epitaxial rise that formed a field of alternating stresses that inhibited dislocation movement and improved the hardness of a given coating. The low value of elastic recovery revealed that the coating had appropriate strength and ductility, as demonstrated by the low value of elastic recovery.

## 4. The Impact of Cutting-Zone Temperatures on Multilayer Coating Properties

During the metal cutting process, about 90% of the mechanical energy is transferred to thermal heat flux, resulting in a severe cutting temperature rise in the cutting zone, as shown in Figure 16 [176,177]. Increased cutting temperature causes excessive wear on the tool rake and flank faces, reducing tool life [178]. Furthermore, induced thermal softening of the tool and workpiece impairs chemical element diffusion and has an impact on surface quality, dimensional accuracy, and functioning of the machined item [179]. As a result, cutting temperature is an important metric for predicting tool wear and cutting performance [180].

According to Nguyen et al. [181], a coating made of (Ti,Al,Cr,Si)N may be thermally oxidized at 1000 °C. Cr_2_O_3_, “Al_2_O_3_”, and “Rutile” (a type of titanium dioxide) oxide phases developed on the coating’s surface under these circumstances. Al_2_O_3_ and Cr_2_O_3_ were largely found in the inner section of the oxide layer, whereas TiO_2_ occupied the outside part. In the oxide layer, the amount of SiO_2_ was negligible because of its limited mobility. The researchers identified a four-sublayer oxide layer structure. The outer layer was created via the incorporation of Si ions into the TiO_2_ phase (the layer was formed due to the diffusion of Ti, Si, and to a lesser extent). 

The Cr_2_O_3_/Al_2_O_3_ with a minor number of Si ions was the next layer, followed by the TiO_2_ phase, which had a low inclusion of Cr, Al, and Si ions in the inner sublayer. Oxygen diffusion caused the formation of the inner layers. The (Ti,Cr,Al)N coating’s thermal stability, oxidation resistance, and tribological characteristics were studied by Xu et al. [182]. This coating was found to have a hardness of between 32 and 33.9 GPa at room temperature. A Cr content of less than 38% prevailed over annealing in an inert environment, resulting in a hardness increase of up to 35 GPa when heated to 700 °C. High Cr concentration in the coatings led to the formation of Cr2N, which resulted in metallic Cr and the spinodal breakdown of the supersaturated c-(Ti,Cr,Al)N supersaturated solid solution when heated. There was also a sign of dispersion hardening. Solid solution (Cr,Ti)_2_N phase was produced due to the high Ti concentration. A considerable drop in hardness was seen when the Cr concentration in AlN approached 41%, which was followed by the quick formation of the hemodynamically stable wurtzite phase during annealing (associated with the dissolution of Cr–N bonds). Since these metals generate stable oxide coatings that inhibit oxidation, the oxidation resistance steadily rose as the Cr and Al amount increased. Thus, Ti_0.18_Al_0.35_Cr_0.47_N coating had the best oxidation resistance when heated to 900 °C, as shown in Figure 17.

In contrast, the low-Cr Ti_0.26_Al_0.48_Cr_0.26_N coating proved to be an excellent choice for cutting tools because of its oxidation resistance and strong hardness at high temperatures. The (Ti,Al)N/CrN coatings were examined for the influence of temperature and found that at 700 °C, the coating displayed an interface-directed spinodal breakdown, and Al atoms went to the closest interface, as shown in Figure 18. An anisotropic interface-directed decomposition mechanism developed as the temperature rose, accompanied by transboundary interdiffusion as a result (with intensive mixing of Cr and Ti). Columnar grains had clearly defined composition gradients at this stage of disintegration, oriented perpendicular to the layers’ paths. At a temperature of 1000 °C, the underlying coating layer structure totally dissolved but was preserved in places that were far from the surface [183,184]. As a result of this, grain production also increased. Interlayer interdiffusion was seen in the (Ti,Al)N/CrN system at 500 °C, whereas hardness decreased at 600 °C, and active oxidative processes were activated at 900 °C on the coating’s surface. At 800–1100 °C, an AlCrTiSiN nano multilayered coating was isothermally annealed by Chen et al. [185] principally consisting of *fcc-* AlCrTiSiN solid solution and some *hcp*-clusters at *fcc*-AlCrTiSiN nanotwin borders. The findings showed that annealing at 1100 °C promotes layered structure disintegration by pinching off hcp sublayers; the transition of fcc AlCrTiSiN solid solution to Cr, Cr_2_N, and *hcp*-AlN particles; and the coarsening of *hcp*-AlN particles, leading to a significant loss in hardness.

### 4.1. Failure Behavior of Coated Tools

The coated tool performance is greatly influenced by residual stresses formed in the film structure [186]. Compressive stresses are thus regarded beneficial for loaded components and tools, as they prevent fracture initiation and propagation in the coating. Higher compressive stresses are linked to longer coated tool life until a particular stress limit is reached, which worsens coating brittleness and hence cutting efficiency [187]. PVD coatings on cutting tools have received widespread acceptance and have emerged as a key means of increasing tool cutting quality. Due to thermoelastic forces and grown-in flaws caused by particles with high kinetic energy during deposition, PVD coatings often have substantial residual stresses in their structure [188].

The amplitude of produced compressive residual strains during film deposition is influenced by process factors such bias voltage [189]. The goal of the research was to figure out how the degree of residual stresses in the PVD film structure affects the wear behavior of coated tools. The coating residual stresses are predicted to have a major impact on attributes such as fatigue, toughness, residual stresses, and adhesion, which are essential for cutting with coated tools. Novel experimental approaches combining with FEM-supported calculations were used to establish these features. Residual stresses are a critical aspect in component longevity and are a component of surface integrity.

Compressive residual stresses are beneficial for tool life because cutting tools have alternating mechanical and high thermal demands. As a result, during the production of cutting tools, it must be guaranteed that the ready-made tools have appropriate compressive residual stress in their subsurface. Some special properties of PVD-coated cutting tools must be examined in this context. The base must be made up of at least two phases: one or more hard materials and a binder substance. Residual stresses are usually solely taken into account for the dominant hard material, while there may be interdependencies between coating and binder residual stresses. The fabricated tool’s ultimate residual stress state is the outcome of a sequence of machining stages, each of which impacts or even produces a new residual stress state. Because the depth of susceptible to influence coating subsurface is restricted to around 10 mm, it is believed that the process step not only alters the current residual stress state but also creates a new subsurface condition.

The facts that each machining step removes material from the surface and the new subsurface only partially overlaps the old one support this. It is important to remember that not only thermal component loads cause a shift in the residual stress state in the direction of tensile stress while making PVD-coated cutting tools. During etching, for example, more binder material is removed than coating grains, resulting in a relaxation of the coating’s compressive stress and a shift towards the tensile stress direction due to mechanical effect. Furthermore, the presence of a coating with a high compressive residual stress causes the substrate stress to shift in the direction of tensile stress. Mechanical sources in addition to thermal ones are responsible for this.

A compensation of the high compressive coating stress occurs in the substrate subsurface due to mechanical causes, resulting in the above-mentioned displacement. High compressive coating stress is necessary to extend tool life duration. There is an upper limit beyond which tool life begins to dwindle anew. However, significant compressive stress in the substrate subsurface is also necessary to prevent cohesive tool damage, which is only possible within specific limitations since coating and substrate residual stresses interact. In this case, finding a balance that combines a lengthy life duration with a reduced chance of cohesive tool damage is critical. These considerations must take into account the effects of surface condition on coating adhesion. The damage behavior of milling inserts is affected by thermal and mechanical loadings in milling operations. Majid Abdoos et al. [190] reported research into the cutting performance and associated coating features of multilayer-thick TiAlN coatings with various residual stress designs. During the deposition process, residual stress was controlled by adjusting the substrate bias voltage. Nano-indentation and scratch tests were used to investigate the influence of residual stress on attributes such as hardness, yield strength, and adhesion. Furthermore, a scanning electron microscope was used to examine the prevalent wear pattern, particularly on the rake face and cutting edge (SEM).

With larger substrate bias voltages and hence higher residual stresses, the findings showed enhanced mechanical characteristics such as hardness and yield strength. Due to a combination of good adherence to the substrate and minimal as-deposited deficiencies, the coating with the lowest compressive residual stress outperformed the other coatings during machining, thereby delaying cutting-edge exposure. Nemetz et al. [191] used the finite element approach to mimic industrial milling operations and obtain information about the underlying damage mechanisms. Two recently published experimental milling settings are used to validate the results. The creation and propagation of so-called comb fractures in planes perpendicular to the cutting edge is aided by these tensile residual stresses, which are damaging to the tool’s efficiency. The insert is constructed of WC-Co hard metal with an average WC grain size of 1 μm and an 8 wt.% co-binder. As a thermal shield, it is covered with a 7 μm thick TiAlN layer. A Johnson–Cook constitutive material model describes the workpiece material, 42CrMO4. A 2D Arbitrary Lagrangian–Eulerian (ALE) technique is used to simulate the milling process. The present study’s key findings are as follows: For 50 cycles of milling, the time-dependent development of fields of temperature and stress in a milling tool was computed in a 3D FE model as a function of milling process parameters. The computed thermal and mechanical stresses were projected to cause localized plastic deformation of the hard metal substrate and, as a result, the accumulation of tensile residual stress in a location near the tool’s rake face’s cutting edge. The suggested modeling approach may be used to forecast the lifetime of metal cutting tools that have failed owing to comb cracking. The fabrication and characterization of superposed structures made of Cr, CrN, and CrAlN layers are described by Tlili et al. [192]. Physical vapor deposition (PVD) was used to deposit CrN/CrAlN and Cr/CrN/CrAlN nano-multilayers onto Si (100) and AISI4140 steel substrates. The Cr, CrN, and CrAlN monolayers were created using a novel PVD coatings technology that corresponded to deposition with various residual stresses.

Tracks of composition and wear scanning electron microscopy, high-resolution transmission electron microscopy, atomic force microscopy, X-ray photoelectron spectroscopy, energy-dispersive X-ray spectroscopy, X-ray diffraction, and a 3D-surface analyzer were used to describe the coating morphologies. Nanoindentation, interferometry, and micro-tribometry were used to study the mechanical characteristics (hardness, residual stresses, and wear) (fretting-wear tests). Multilayer coatings appear to be largely made up of nanocrystalline, according to observations. The number of residual stresses in the films has had a significant impact on all physicochemical and mechanical parameters as well as wear behavior. As a result, the coating with moderate residual stresses has a better wear behavior than the coating with larger residual stresses. Friction interaction between coated samples and alumina balls reveals a wide range of wear processes as well. Plastic deformation, tiny microcracking, and micro spallation were all involved in the abrasive wear of the coatings. 

### 4.2. Residual Stresses Induced in Coating Process

Residual stresses present after the coating process depend heavily on the method and properties of the coating process, on the coating material, and lastly on the substrate. Three different processes have been identified as the source of majority of residual stress: ceramic phase transformation during cooling process, cooling of molten spray, and cooling of the whole coating system [193]. These processes are a factor in methods relying on a difference in temperatures (spray coating or vapor depositions, for example) and are not present in methods using electrolytic or electrophoretic processes to deposit the coating even though stresses produced by mismatch of a lattice constant will be present no matter the coating method. The severity of the residual stresses depends on the temperature during the coating process, cooling speed, and the thickness of the generated layer as well [194]. To reduce the residual stress, several methods have been developed, mostly focusing on temperature control and strain compatibility at layer interfaces [195,196]. Generally, a better matching coating material to the substrate needs to be used, but where that is not possible, there are several solutions. One of the approaches is a multi-layered coating. This can allow for a deposition of a more heat-resistant material first, reducing the stress caused by temperature change in the final layer. This combination can take the form of, for example, heat-resistant ceramic and a metallic coat [197]. While this method drastically reduces the thermal residual stress, it is important to take into account the mechanical properties of the different coatings as well because heat-resistant material might not have the best properties required of the final coating, such as vibration resistance or thermal/electric conductivity or isolation. A slightly different approach during layer deposition is to reduce the thermal expansion mismatch by using coatings with graded properties [193,198]. In this process, the physical properties change in a stepwise manner due to change in composition or morphology [199]. In this way, it is possible to transition from properties of the coating that will provide better adhesion with the substrate but do not possess the qualities of the desired top layer to more ideal top layer of the coating without the need to trade adhesion compatibility for desired hardness or resistance and vice versa. The third option is the combination of the previous two, where a multi-layered coating is improved by inserting a graded interlayer. This technique has the potential to provide the best possible result, but it has drawbacks, such as the technological requirements on the whole process, precision of material selection, and the final thickness of the whole coating, which can increase dramatically when including more and more layers. This large number of different methods combined into one increases the number of failure states, which increases the difficulty of the whole process [200].

## 5. Future Challenges for Hard Coatings

Since the introduction of hard coatings in the late 1960s, industry aspirations to further enhance these coatings have been fueled by the push to enhance the productivity and competitiveness of manufacturing processes and, later, by the desire to reduce the use of coolants and lubricants and to save resources by extending the life span of tools. This rapid and ongoing success of hard coatings contributes significantly to the sustainable development goals (SDGs), for example, by lowering the energy consumption of cutting procedures. While this motivation will undoubtedly continue, the already existing basic scientific foundation is being exploited to provide more and more functionality to these coatings. However, in the future, the hard coatings community, i.e., both academic and industry, will have to greatly strengthen their efforts to meet the SDGs, as will any other production-related group, pushed by the appropriate government laws. Some of these prospective difficulties will be addressed in the following paragraphs.

Multi-functionality coatings

There have previously been significant attempts to create multifunctional hard coatings in order to achieve a long tool lifespan, but only a handful of these have found their way into industrial applications. Self-hardening (for example, age hardening of Ti_1-x_Al_x_N-based coatings) has been a focus of study for over two decades [201]. In recent decades, complex coating topologies for damage-tolerant coatings capable of stopping fractures have been proposed. The obvious next step is the development of self-healing coatings, which is currently in its early stages. Surface self-organization processes during tribological sliding contacts are seen as a potential technique for not only self-healing of surface fractures but also for adding self-lubrication functions, making the coated product more environmentally friendly [202]. Furthermore, thermal management abilities have grown in popularity in recent years, as the conventionally required low thermal conductivity of hard coatings may result in high peak temperatures in the contact with the chip in cutting and distinct thermal gradients, implying that a tailored thermal conductivity allowing heat isolation along the coating thickness and heat distribution within the coating plane may be more advantageous [203]. Autonomously self-healing hard coatings are a novel technique in which microstructural changes are tracked by in situ measurements of relevant characteristics, such as electrical sheet resistance, to follow oxidation progress. All these characteristics should be combined in an original design for a future hard coating for metal-cutting applications [204].

Simulated coatings and deposition processes

Simulation-guided or at least -helped design of hard coatings with improved performance is already a well-established approach that aids in resource conservation. The existing technique, however, has to be encouraged and expanded further. There is now a significant gap between turn-key solutions for typical applications, which can be handled by semi-skilled individuals, and unique coating concepts, which require specially qualified academics. To bridge this gap and enable the rapid and easy construction, correction, and optimization of deposition processes without the immediate requirement for highly educated experts, the development of related technologies [205] and, as a result, expert systems for the individual deposition processes will be required. Adding multifunctional features to more complicated hard coatings will require the digitalization of our traditional coating development processes to profit from the ever-increasing stream of accessible data. Modern data management systems, for example, based on the FAIR Guiding Principles [206] (where FAIRness of data implies that it is findable, accessible, interoperable, and reusable), could enable machine learning in the future from a wealth of theoretical and experimental data that humans would never be able to overlook.

Energy-efficient deposition techniques

Coating deposition machines use a huge amount of energy to manufacture a relatively little amount of coating material. Normalization of the collected data in terms of coating thickness and number of tools per coating batch enables comparison of the efficiency of various deposition procedures (energy consumption and material fluxes in PVD and CVD processes). In addition to visualizing energy consumption and material fluxes, the developers suggest considering ways to improve the energy efficiency of deposition processes with high energy consumption (e.g., heating substrates or running power supplies for evaporation or sputtering) in next-generation coating facilities. For example, the utilization of heat-exchange modules for waste heat recovery offers previously untapped opportunities to improve the energy efficiency of deposition systems and so contribute to environmental sustainability.

Coating thermal and mechanical properties.

Future research should focus on studying the impact of tool coating wear and failure progression on cutting temperature and tool life across the whole metal-cutting process with coated tools. Based on genuine experimental observations, the assumption of boundary conditions in prediction models of cutting temperature should be updated to be compatible with the actual cutting process. Though certain cutting temperature measuring techniques with uncoated tools are applicable to coated tools, future research should focus on developing and improving accurate cutting temperature measurement techniques with rapid reaction times. There have been few investigations into the temperature monitoring and control of coated smart cooling tools. Cutting temperature reductions are more noticeable with coated smart cutting tools than with uncoated smart cutting tools due to the coating’s superior effect and internal cooling effects. Recent studies have focused mostly on the coating mechanical characteristics of micro-milling tools. There has been little research into the impact of coating thickness or thermal characteristics on micro-milling temperatures, including analytical models and trials. The tool edge radius increased as the coating layer grew, impacting the micro-milling temperature. However, the influencing mechanism remains unknown. The proper coated micro-milling tools must be built with both coating thermal and mechanical qualities in consideration.

## 6. Conclusions

This paper presents a review of the current state-of-the-art in hard coatings for metal cutting applications, focusing on widely used and well-established hard coatings used in the metal cutting industry and considering the similarities and differences between coatings grown using PVD and CVD techniques. Economic and environmental considerations in the machining industry result in an ongoing increase in cutting velocities and feed rates as well as a reduction in the use of coolants and lubricants, with the concurrent demand for longer tool lifetimes, necessitating continuous further development of the applied hard coatings.

Historically, strong wear resistance was frequently associated with great hardness and perhaps good thermal and oxidative stability. Additional qualities are now necessary to satisfy ever-increasing demands and withstand the severe circumstances that cutting instruments are subjected to. Modern hard coatings must be multifunctional and have qualities such as enhanced breaking resistance or advanced heat management, which are generally achieved through complex coating designs. Modern nanocoatings demonstrate greater adhesion and increased wear resistance, resulting in longer tool life. This demonstrates the coatings’ tremendous promise in a variety of milling applications, particularly the milling of difficult-to-machine materials.

Recent studies have also concentrated on nanostructured and nanolayered coatings. These have been tested and found to have high tool-life values, rivalling those of more recent nanocoatings, with very satisfactory results.

In terms of tool wear processes and patterns, for milling tools, the most common wear pattern is the creation and spread of cracks, which is produced by thermal fatigue (high machining temperatures). Recent research is still heavily focused on the causes and patterns of tool wear that are experienced by coated tools because this directly affects the tool’s lifespan. Studies on coating wear might benefit new coating development and process optimization. According to recent research, coatings can be used longer if they are applied using these procedures, providing excellent outcomes in terms of both surface finish and overall process efficiency.

## Figures and Tables

**Figure 1 materials-15-05633-f001:**
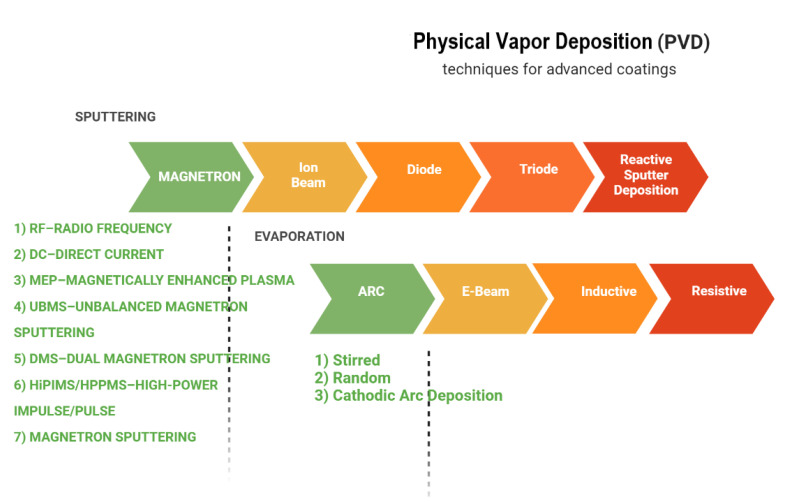
Physical vapor deposition (PVD) methods used for improved coatings.

**Figure 2 materials-15-05633-f002:**
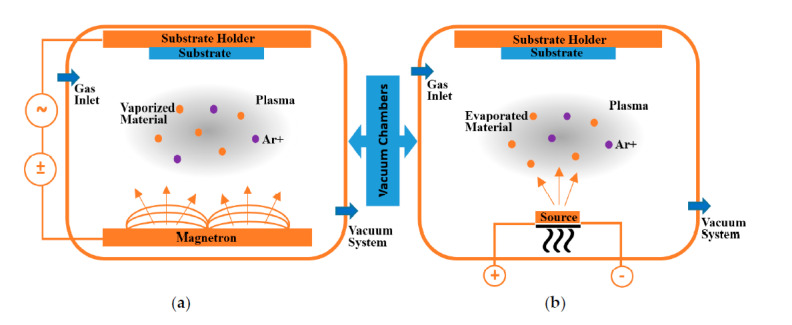
Schematic drawing of two conventional PVD processes, (**a**) sputtering and (**b**) evaporating, using ionized Argon (Ar+) gas [32]. Reproduced with permission from Baptista, A, Coatings; Published by MDPI, 2018.

**Figure 3 materials-15-05633-f003:**
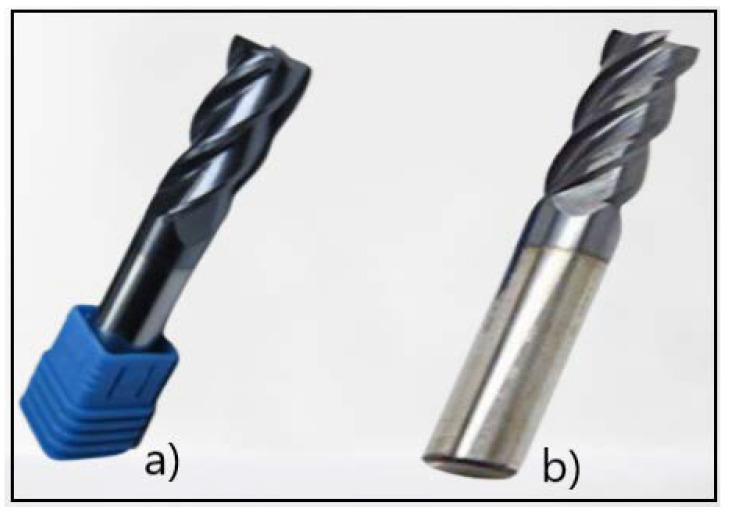
Uncoated carbide tool (**a**) and TiAlN-coated carbide tool (**b**) [82]. Reproduced with permission from Elsevier.

**Figure 4 materials-15-05633-f004:**
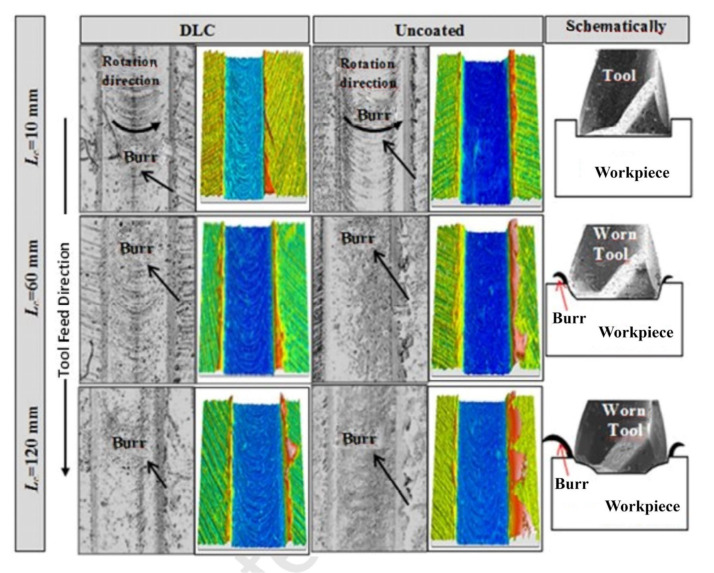
Burr formation occurring in cutting process (f = 5 μm/flute, ap = 0.15 mm) [88]. Reproduced with permission from Elsevier.

**Figure 5 materials-15-05633-f005:**
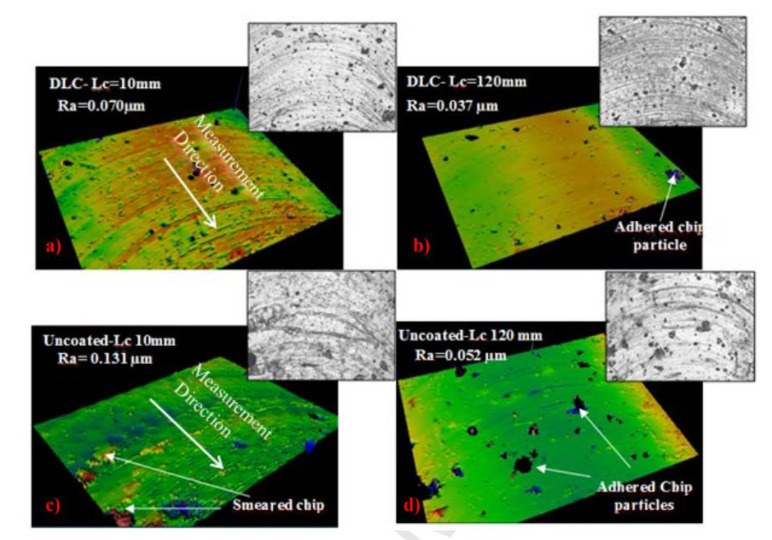
Three-dimensional topographic images of the machined surfaces (f = 2.5 μm/flute, ap = 0.15 mm) after cutting process of 10 mm [88]. Reproduced with permission from Elsevier. (**a**) DLC (Lc = 10 mm) (**b**) DLC (Lc = 120 mm) (**c**) Uncoated (Lc = 10 mm) (**d**) Uncoated (Lc = 120 mm).

**Figure 6 materials-15-05633-f006:**
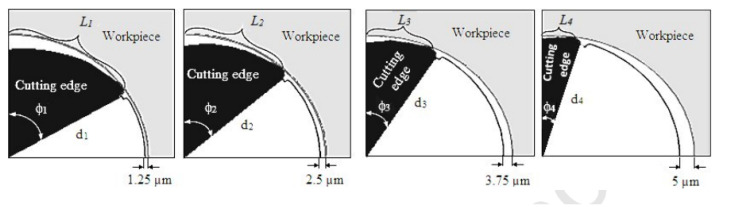
Schematic drawing of chip formation process depending on angular position of tool for different feed rates (ϕ1 > ϕ2 > ϕ3 > ϕ4 and L1 > L2 > L3 > L4) [88]. Reproduced with permission from Elsevier.

**Figure 7 materials-15-05633-f007:**
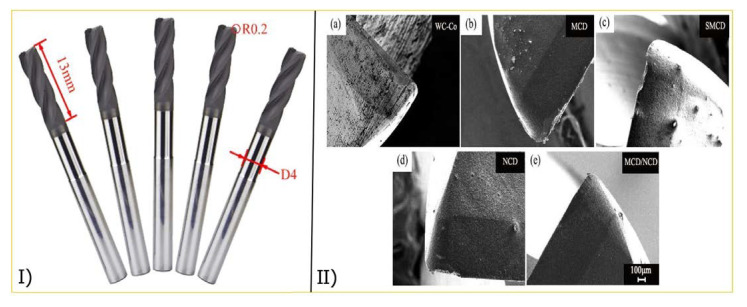
(**I**) Diamond-coated corner radius end milling tools, (**II**) tool wear morphologies of different kinds of milling tools: (**a**) WC-Co; (**b**) MCD; (**c**) SMCD; (**d**) NCD; (**e**) MCD/NCD [90]. Reproduced with permission from Elsevier.

**Figure 8 materials-15-05633-f008:**
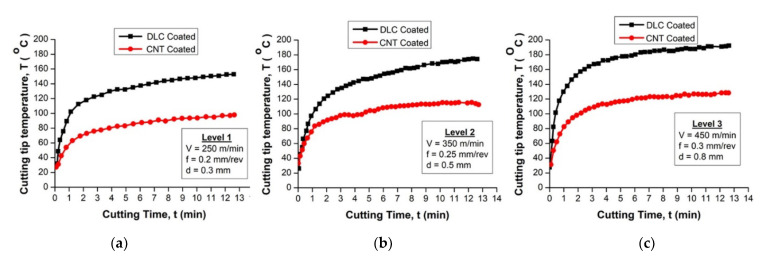
Temperature results at (**a**) level 1, (**b**) level 2, and (**c**) level 3 [97]. Reproduced with permission from Elsevier.

**Figure 9 materials-15-05633-f009:**
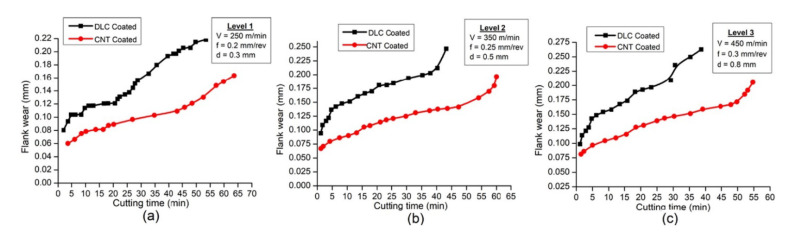
Flank wears results at (**a**) level 1, (**b**) level 2, and (**c**) level 3 of cutting conditions [97]. Reproduced with permission from Elsevier.

**Figure 10 materials-15-05633-f010:**
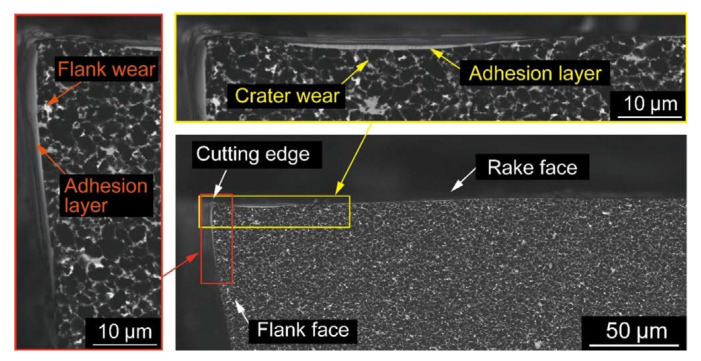
Cross-section observation of the CBN cutting tool after cutting for 25 m [142]. Reproduced with from permission from Elsevier.

**Figure 11 materials-15-05633-f011:**
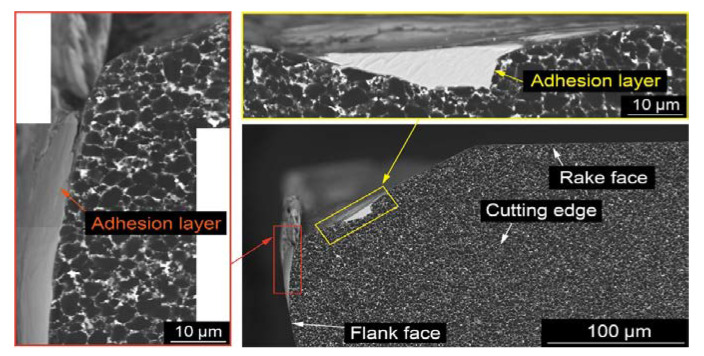
Cross-section observation of the CBN cutting tool after cutting for 200 m [142]. Reproduced with permission from Elsevier.

**Figure 12 materials-15-05633-f012:**
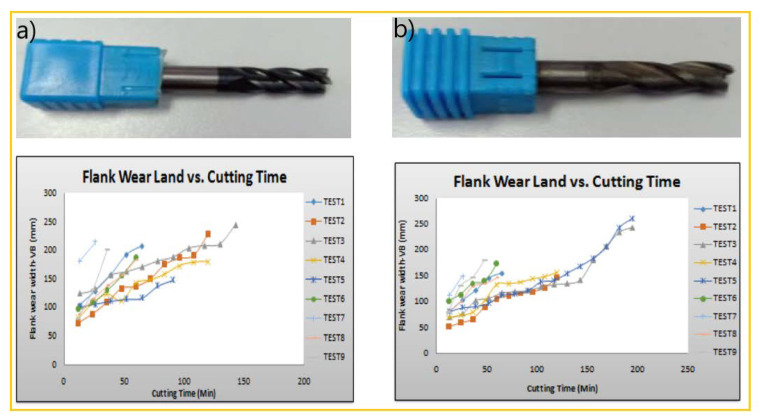
(**a**) Uncoated HSS Flank Wear Land vs. Cutting Time; (**b**) Coated HSS Flank Wear Land vs. Cutting Time [143]. Reproduced under CC BY license.

**Figure 13 materials-15-05633-f013:**
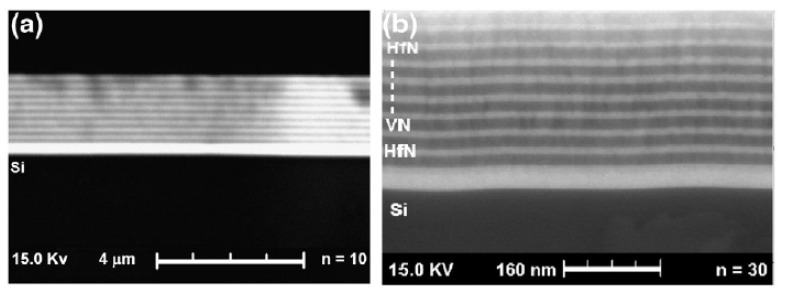
SEM micrograph of HfN/VN multilayer system: (**a**) coating deposited with n = 10, bilayer periods (Λ = 120 nm); (**b**) coating deposited with n = 30, bilayer periods (Λ = 40 nm) [154]. Reproduced with permission from Elsevier.

**Figure 14 materials-15-05633-f014:**
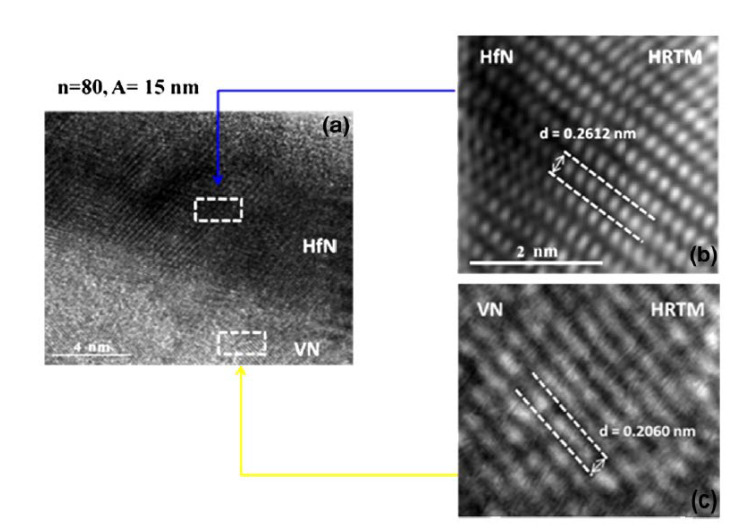
(**a**) TEM image of HfN/VN multilayer coatings with n = 80, bilayer periods (A = 15 nm) atomic microstructures of HfN (**b**) and VN (**c**) layers [154]. Reproduced with permission from Elsevier.

**Figure 15 materials-15-05633-f015:**
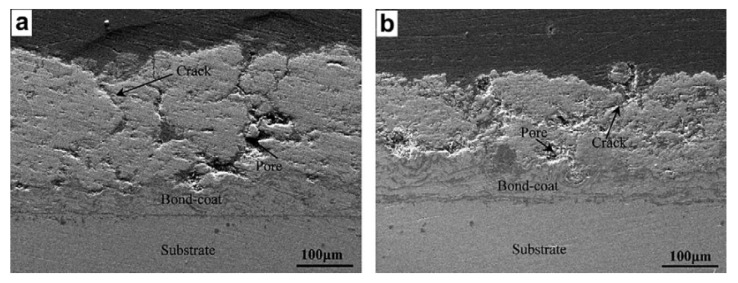
The section image the as-sprayed (**a**) conventional 8YSZ TBC and (**b**) nanostructured 8YSZ TBC [157]. Reproduced with permission from Elsevier.

**Figure 16 materials-15-05633-f016:**
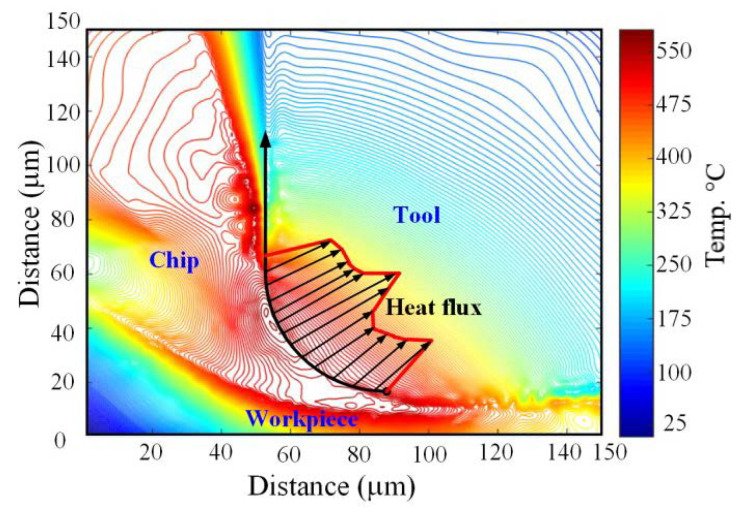
Microscale modeling of cutting temperature distribution at the tool–chip contact in the cutting zone [177].

**Figure 17 materials-15-05633-f017:**
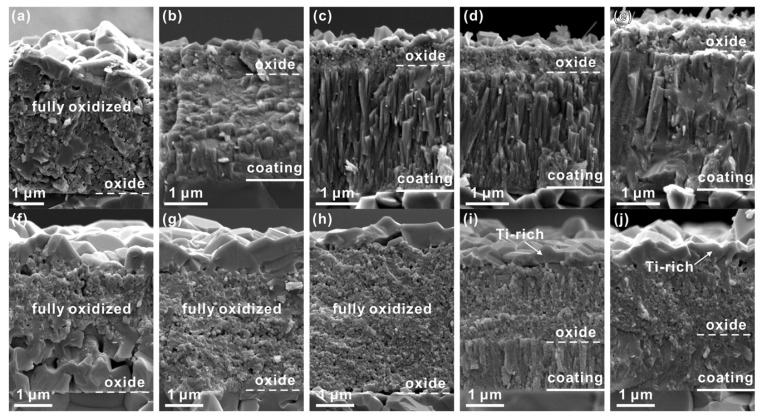
SEM fracture cross-sections of (**a**,**f**) Ti_0.46_Al_0.54_N, (**b**,**g**) Ti_0.35_Al_0.42_Cr_0.23_N, (**c**,**h**) Ti_0.29_Al_0.36_Cr_0.35_N, (**d**,**i**) Ti_0.26_Al_0.33_Cr_0.41_N, and (**e**,**j**) Ti_0.24_Al_0.__29_Cr_0.47_N coatings with an Al/(Ti + Al) ratio of ~0.55 after oxidation at (**a**,**e**) 850 °C and (**f**,**j**) 900 °C for 20 h [182]. Reproduced with from permission Elsevier.

**Figure 18 materials-15-05633-f018:**
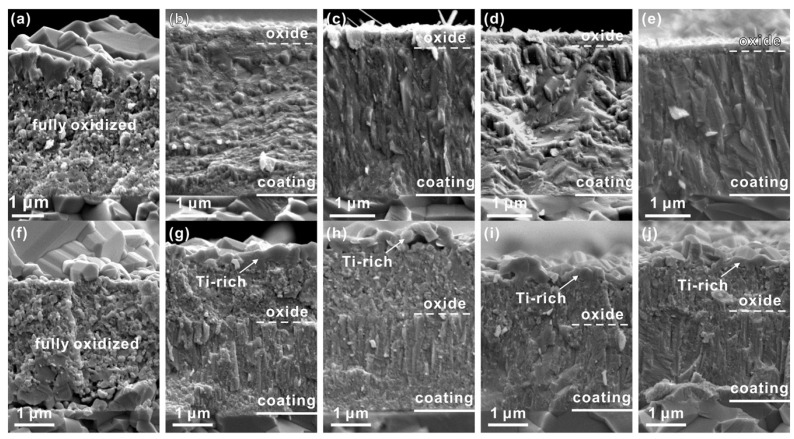
SEM fracture cross-sections of (**a**,**f**) Ti_0.34_Al_0.66_N, (**b**,**g**) Ti_0.26_Al_0.48_Cr_0.26_N, (**c**,**h**) Ti_0.22_Al_0.40_Cr_0.38_N, (**d**,**i**) Ti_0.19_Al_0.38_Cr_0.43_N, and (**e**,**j**) Ti_0.18_Al_0.35_Cr_0.47_N coatings with an Al/(Ti + Al) ratio of ~0.65 after oxidation at (**a**,**e**) 850 °C and (**f**,**j**) 900 °C for 20 h [182]. Reproduced with permission from Elsevier.

**Table 1 materials-15-05633-t001:** Summary of coating processes and their advantages and limitations.

Deposition Process	Source	Substrate Material	Coating Thickness (µm)	Advantages	Disadvantages	Refs.
PVD	Physical	AISI M2 steel, SS, glass, Si, potassium bromide (KBr)-carbon-Au-Al, Ag-Au-Cu-Al	1.2–6.3, 5, 0.2, 0.2, 0.1, 0.1	Corrosion and wear resistance/thin film deposition is possible/adjustable mechanical, corrosion, and aesthetic properties	Requires a high vacuum/corrosion resistance is affected by abrasion/degradation control is challenging for polymer deposition applications	[56,57,58,59,60]
CVD	Chemical	Glass, Si, Si, Kleenex, Ni-Co-Fe	0.05–0.2, 0.2–0.6, 0.04–0.1–16	Corrosion and wear resistance/deposition of various types of materials with different microstructures/works with low and atmospheric pressures	Requires ultra-high vacuum/requires heat-resistant substrates/small amount of coating materials waste	[61,62,63,64]
ELD	Electrochemical	Steel, carbon steel, mild steel, Cu	50–200, 10–70	Decorative and low corrosion/wear applications/high-temperature applications	Works for conductive substrates	[43,46]
EPD	Electrochemical	AISI 316L SS, AISI 304 SS, AISI 316L SS, Ti-6Al-4V alloy (TC4), Aramid-carbon-cellulose fibers composite	7, 1–6	Various kinds of selective, graded material, and porous structure depositions/biomedical applications/wear resistant	Works for conductive substrates	[49,52]
Plasma spray	Thermal	SS, steel, AISI 4140 steel	0.5–1	High corrosion and wear resistance/high substrate adherence/surface modification of engineering polymers, rubbers, metals, and fibers/anti-stick coatings	A low-temperature process that is mostly used for materials that cannot perform reactions in atmospheric pressure to modify the surface of the substrate/requires a heat source	[65,66]
HVOF	Thermal	Ti-6Al-4V, Inconel 738 metal, AISI 4340 SS	70, 100	High density of coating layer and well substrate adherence/works for non-conductive substrates/corrosion and wear resistance	Requires a small range of powder size (5–60 µm) with a narrow size distribution/numerous process variable to change the coating structure/requires a heat source	[67,68]
Cold Spray	Physical	Ti-6Al-4V, Al 6061-T6, Al 6061	100–1000, 40–300	Simple and cheap method compared to the other thermal spray methods	Limited operation range/mostly used for soft and hard metal substrates/low efficiency and reliability due to low temperatures/not useful extremely harsh environments	[69,70]
Warm Spray	Physical	316L SS, steel, steel, carbon steel	400–1000, 400, 300	Applicable to materials with sensitivity to oxidization at high temperatures or heat-sensitive materials	Impurity complications/not useful extremely harsh environments	[71,72]
Arc Wire Spray	Thermal	Carbon steel, SUS 304	1000	Internal surface coatings such as engine blocks/wear and corrosion resistant	Limited to conductive wires and materials as the coating layer	[73,74]

**Table 2 materials-15-05633-t002:** Results of comprehensive research into the machining performance of several steels in hard turning using tools with various AlTiN and AlTiSiN coating materials under various machining settings.

Ref.	Materials Studied	Cutting Tool	Principles of Cutting	Machining Responses Evaluated	Results
[106]	AISI 4340	TiC-coated carbide	Cutting speed, feed, depth of cut, cutting environment	Cutting force, chip morphology	Good machinability is observed with cutting velocity more than 100 m/min. Wide groove-type chip breaking insert outperformed other inserts regarding different machining characteristics.
[107]	AISI 4340	Multilayer CVD-TiN/Al_2_O_3_/TiCN and monolayer PVD-TiCN coated carbide	Cutting speed, feed, depth of cut	Surface roughness, flank wear, cutting temperature, material removal rate	Monolayer PVD-coated carbide has outperformed multilayer CVD-coated carbide tool almost for all cutting conditions except at a high level. MRR was found to be higher for multilayer CVD-coated carbide tool than for PVD-coated carbide
[108]	AISI 4340	Single-layer TiAlN, multilayer MT-TiCN/Al_2_O_3_/TiN-coated carbide	Cutting speed, feed, depth of cut	Crater wear, flank wear, cutting force	Abrasion, diffusion, and adhesion were the dominant wear mechanisms. Multilayer TiCN/Al_2_O_3_/TiN carbide insert was observed to be more effective in producing better tool life. Speed was the dominating parameter for tool life
[109]	AISI 4340	PVD-TiAlN and TiCN/Al_2_O_3_/TiN multilayer coated carbide	Cutting speed, feed, depth of cut	Tool life	PVD-TiAlN-coated insert performed better than CVD-TiCN/Al_2_O_3_/TiN-coated inserts.
[110]	AISI 4340	TiAlN/TiN, TiCN/Al_2_O_3_ multilayer-coated carbide and TiAlN/AlCrN-coated cermet	Cutting speed, feed, depth of cut	Chip morphology, flank wear, crater wear	Carbide tools performed better than cermet, and TiAlN/TiN-coated carbide outperformed TiCN-Al_2_O_3_-coated tool
[111]	AISI 4340	TiN + TiCN + Al_2_O_3_ multilayer-coated carbide	Cutting speed, feed, depth of cut, machining time	Surface roughness, force, chip morphology, tool wear	With a lower range of machining parameters, better machining characteristics can be achieved.
[112]	AISI 4340	TiN/TiCN/Al_2_O_3_ multilayer-coated carbide	Cutting speed, feed, depth of cut, MQL pulsating time	Tool failure, surface roughness, surface topology, chip morphology	Pulsating MQL showed better results in terms of surface roughness and flank wear. The chip reduction coefficient is highly affected by the feed rate
[113]	AISI 4340	TiC + TiCN + Al_2_O_3_ multilayer-coated carbide	Cutting speed, feed, depth of cut	Surface roughness, machining forces, specific cutting force, power, flank wear	Higher feed value was favorable for specific cutting force, and higher speed was favorable for surface finish.
[114]	AISI 4340	TiAlN and TiCN + Al_2_O_3_ + TiN multilayer-coated carbide	Cutting speed, feed, depth of cut, w/p material hardness	Surface roughness, machining forces, flank wear, chip morphology	Better tool life was observed for TiCN + Al_2_O_3_ + TiN-coated carbide tool. Least roughness was observed for TiAlN-coated tool
[115]	AISI 4340	TiCN + Al_2_O_3_+ TiN multilayer-coated carbide	Cutting speed, feed, depth of cut	Surface roughness	Feed was the most influential parameter for surface roughness.
[116]	AISI 4340	TiN/TiCN/Al_2_O_3_/TiN multilayer-coated carbide	Cutting speed, feed, depth of cut	Surface roughness, chip morphology, flank wear	Speed and feed influenced both the surface roughness and flank wear
[117]	AISI 4340	TiN/TiCN/Al_2_O_3_/TiN and TiN/TiCN/Al_2_O_3_/ZrCN multilayer-coated carbide	Cutting speed, feed, depth of cut, cutting time	Cutting forces, surface roughness, chip morphology, flank wear	Multilayer coated carbide performed better than uncoated one. From economic analysis, multilayer coated carbide was also favorable
[118]	16MnCrS5 stee	TiC-coated carbide	Cutting speed, feed, depth of cut, cutting environment	Main cutting force, surface roughness, flank wear, chip morphology	Tool wear rate in wet condition is found to be lesser than dry condition. Cutting velocity is found to have an influence on the main cutting force than the machining environment.
[118]	AISI 4140	Coated carbide inserts (TiN, TiN +TiAlN +TiN, TiN +TiCN + Al_2_O_3_, TiN + TiCN +Al_2_O_3_ + TiN)	Cutting speed, feed, depth of cut	Cutting force, crater wear, flank wear, cutting temperature	Multilayer-coated (TiN-TiCN- Al_2_O_3_-TiN) inserts achieve the longest tool life. The minimum temperature was observed with the outer layer coating having Al_2_O_3._
[119]	C45 steel	TiN-coated carbide	Spindle speed, feed, depth of cut, nose radius	Surface roughness	Nose radius highly influenced the surface roughness followed by feed rate, spindle speed and depth of cut.
[120]	AISI D2	Coated carbide inserts (TiN, TiAlN, and TiCN)	Cutting speed, feed, depth of cut, tool coating material	Surface roughness, flank wear, material removal rate	The coating material of the tool was the dominant parameter for the responses. TiAlN outperformed the other two coatings.
[121]	AISI D2	TiN + TiCN+ Al_2_O_3_+ TiN multilayer-coated carbide	Cutting speed, feed, depth of cut, w/p material hardness, machining time	Surface roughness, flank wear	Coated carbide tool outperformed the uncoated tool in every aspect
[122]	AISI D2	TiN + TiCN + Al_2_O_3_ multilayer-coated carbide	Cutting speed, feed, depth of cut	Surface roughness, chip morphology, chip-tool interface temperature, flank wear	Coated carbide outperformed uncoated carbide in every aspect.
[123]	Wrought super alloys	TiAlN-coated carbide	Cutting speed, feed, depth of cut	Flank wear, surface roughness	Speed was the dominating parameter for tool life.
[124]	16MnCrS5 steel	TiAlN-coated carbide	Cutting speed, feed, depth of cut	Cutting force, flank wear, surface roughness	Brushed inserts performed better than ground inserts in every aspect.
[125]	Inconel 718	TiAlN-coated carbide	Cutting speed, feed, depth of cut	Cutting force, friction, cutting temperature, tool wear	PVD-TiAlN coated insert with a coating thickness of 1 micron performed better than other inserts.
[126]	AISI 52,100	AlCrN- and AlTiN-coated carbide	Cutting speed, feed, depth of cut	Flank wear, crater wear, friction, chip sliding velocity	Chip sliding velocity during machining increased with the increase in cutting speed. AlTiN-coated tools exhibited superior antioxidation, anti-abrasive, and anti-adhesive behavior as compared to AlCrN-coated and uncoated cutting tools
[127]	AISI 52,100	TiAlxN-coated carbide	Cutting speed, feed, depth of cut	Cutting forces, flank wear, surface roughness	Speed influenced the flank wear much com- pared to other parameters. Carbide inserts with coating thickness 12 micron showed better result compared to 8 mm coating thickness
[128]	AISI 52,100	TiCN + Al_2_O_3_ + TiN multilayer-coated carbide	Cutting speed, feed, depth of cut	Surface roughness, cutting force	Feed was the dominating parameter for surface roughness, and the depth of cut affected the cutting force.
[129]	AISI 52,100	TiN-TiCN-Al_2_O_3_-TiN multilayer-coated carbide	Cutting speed, feed, depth of cut	Surface roughness, microhardness	Feed rate was the dominant factor affecting the surface roughness, whereas the cutting speed was the dominant factor affecting the micro hardness.
[130]	HSS steel	PCBN-, TiN-coated ceramic and TiC/TiCN/Al_2_O_3_/TiN multilayer-coated carbide	Cutting speed, feed, depth of cut	Chip morphology, flank wear	The mixed alumina ceramic and coated carbide tool performed better than the CBN tool.
[131]	Inconel 825	TiN/TiCN/Al_2_O_3_/ZrCN multilayer-coated carbide	Cutting speed, feed, depth of cut, cutting time	Cutting force, temperature, apparent coefficient of friction, chip morphology, tool chip contact length, chip microhardness	Better chip characteristics are observed with coated tools compared to the uncoated tool. Cutting force and apparent coefficient of friction were reduced with the coated tool.
[132]	SAE 6150	Al_2_O_3_ + TiCN multilayer-coated carbide	Cutting speed, feed, depth of cut, nose radius	Surface roughness, cutting force	Uncoated tool performed better compared to coated tool in terms of machining responses
[133]	nickel-based super alloy GH4169	AlTiN-coated carbide	Cutting speed, feed, depth of cut	Crater wear, surface roughness, machined surface morphology	Surface roughness was highly influenced by feed rate. However, machined surface morphology is highly influenced by both speed and feed.
[134]	Super-duplex stainless steel	AlTiN- and AlCrN-coated carbide	Cutting speed, feed, depth of cut	Crater wear, flank wear, chip morphology, machined surface morphology	AlTiN insert showed longer tool life, better surface finish, and smaller chip thickness when compared to AlCrN-coated and uncoated inserts.
[135]	SS 304 steel	TiAlN/TiN multilayer-coated carbide	Cutting speed, feed, depth of cut	Crater wear, flank wear, chip morphology, surface roughness	Coated carbide outperformed uncoated carbide in all respect.
[136]	AISI D3	TiN-, Latuma-, AlCrN-coated carbide	Cutting speed, feed, depth of cut	Residual stress, cutting temperature, tool wear, surface integrity	The depth of cut and workpiece hardness influenced the surface roughness. Better tool life was observed with Latuma-coated inserts
[137]	AISI 4140	TiC-coated carbide	Cutting speed, feed, depth of cut	Surface roughness	Surface roughness was significantly affected by feed rate.
[138]	AISI 1040	TiCN-Al_2_O_3_-TiN multilayer-coated carbide	Cutting speed, feed, depth of cut, w/p material hardness	Surface roughness, sound level, power consumption	Feed was the notable parameter for roughness, whereas depth of cut was significant for sound level and power consumption.

## Data Availability

Not applicable.

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
