# Peer review of "Characterization and Evaluation of Engineered Coating Techniques for Different Cutting Tools—Review"

_materials, 2022, doi:10.3390/ma15165633_

Round 1

Reviewer 1 Report

The key following comments need to addressed by the authors for improving the standards with high quality,

  1. Authors need to conduct in-depth critical review in this review paper: Say example: TiO2 nano-coated tool had a 16% longer lifetime than uncoated tool. Why the coating enhances the tool life explain the physics of a process.
  2. It is necessary to highlight the coating methods (plasma spray coating, electrodeposition coating, laser coating, CVD, PVD etc.) and explain its advantages and limitation. Surface preparation methods also need to be explained.  
  3. Authors need to highlight the suitable coating material (say example, Ti coated on HSS, TiC coating on MS), with coating thickness on a base metal and ensure possible tool life with machining conditions (speed, feed etc.).
  4. Authors need to report the units of cutting depth, cutting velocity, and feed rate etc. in line 121-122. Please note that, the details of workpiece and tool material need to be provided.
  5. To ensure better adherence and long tool life, thickness of each coating layer need to be well described.
  6. It’s better to consolidate the base material (AISI 4340, AISI 4140) with different coating materials [62-71] and predict the tool life and recommend the best coating material with details of coating (hardness, coating thickness, coating method, cutting conditions: cutting speed, feed rate and depth of cut). Line 646-647 there are so many typo errors, please check errors.
  7. In Line 707-708, why the Co matrix, TiC was slowly oxidized but did not breakdown or dissolve.
  8. Inter-crystalline network layer rich in Ti, Zr, and Ce encircled the Al2O3 colony in the plasma-treated nanocomposite powder, what is the advantage of three-dimensional network structure.
  9. Methods that inhibit the adhesive breakage and produced strong interlayer bonding between substrate and coating material.
  10. Briefly highlight the residual stresses induced in coating and its negative effect on properties (hardness, corrosion, and wear properties) and methods to reduce the residual stresses need to be explained in detail.

Reviewer 2 Report

The review deals with Characterization and Evaluation of Optimally Engineered
Coatings for Cutting Tools - Review. 
According to the reviewer, the paper is worth publishing at Materials Journal, 
but corrections are needed and then the paper can be accepted for publication in the journal.
While the authors have made considerable research effort, 
the presentation of the paper and the results must be proved. 
Additionally make the following corrections to the manuscript:

Comment 1
Lines 14 - 15: Drilling is a significant machining condition used by the
majority of industrial businesses for connection activities. 

Lines 45 - 46: The workpiece remains motionless while the tool rotates throughout the milling 
operation. 

The authors should explain: drilling or milling.

Comment 2
Lines 58 - 59
Martha,
Kishore, and colleagues [4] investigated

The authors should replace

Martha et al. [4] investigated

Comment 3
The authors must format the paper according to the journal's instructions. Extended text editing.

Line 68
The word "minimizing" has not the same text font size with the rest of the text.  

Line 71
than the base. [6]. The
The authors should replace (delete a .)
than the base [6]. The

Lines 74 - 75
chemical vapor deposition (CVD) or
physical vapor deposition (PVD). 

The authors should replace
Chemical Vapor Deposition (CVD) or
Physical Vapor Deposition (PVD). 

Line 80
molecular beam epitaxy (MBE),
The authors should replace
Molecular Beam Epitaxy (MBE),

Line 82
to CVD, Because of the materials
The authors should replace
to CVD, because of the materials

Line 88
[10]. Silva et al[11] investigated
The authors should replace (insert a space)
[10]. Silva et al [11] investigated

Line 91
The word "utilized" has not the same text font size with the rest of the text.  

Line 93
The word "utilized" has not the same text font size with the rest of the text.  

Line 95
The word "substrate" has not the same text font size with the rest of the text. 

Line 110
Santhanakrishnan et al.[15] used
The authors should replace (insert a space)
Santhanakrishnan et al. [15] used

Line 119
Swain, Mohapatra et al. 
The authors should replace
Swain et al.

Lines 122 - 123
The authors must insert the units (values: 0.2, 70 and 0.10).

Line 145
by Rodriguez-Barrero, S., et al [17].
The authors should replace
by Rodriguez-Barrero et al. [17].

Line 146
steel 42CrMo4, which
The authors should replace (4: normal)
steel 42CrMo4, which

The authors should replace (paper) 
Fig. replace Figure

Line 209
In the duplex stainless-steel face GX2CrNiMoN26-7-4 DSS. 
Something is missing. The authors must rephrase.

Line 215
brittl.
The authors should check the paper for typographical and spelling errors.

Line 217
than CVD coatings. and was con- 
The authors should check the paper for typographical and spelling errors.

Lines 224 - 225
using filtered arc depo- 
sition, which is a newer deposition process (FAD).
The authors should replace
using Filtered Arc Depo- 
sition (FAD), which is a newer deposition process.

Line 237
from chen et al [26]. The
The authors should replace
from Mei et al. [26]. The

Line 241
boron doping Cutting performance
The authors should check the paper for typographical and spelling errors.

Comment 4
Figure 1
The authors should move the 1b and 1c (inside the Figure)

Comment 5
Line 272
from Baptista, A, Coatings; Published
The authors should replace
from Baptista et al., Coatings; Published

Comment 6
The authors should insert a Section "CVD Coatings".

Comment 7
Line 317
The "[36]" has not the same text font size with the rest of the text. 

Line 371
The word "utilization" has not the same text font size with the rest of the text. 

Line 374
W/mK)[40].
The authors should replace (insert a space)
W/mK) [40].

Line 375
Pazhanivel et al.[41]. they were found
The authors should replace
Pazhanivel et al [41]. They were found

Line 379
by Hirata et al.[42] According
The authors should replace
by Hirata et al. [42]. According

Line 397
by Chenrayan, V., et al [46] looked
The authors should replace
by Chenrayan et al. [46] looked

Line 434
Level 3[46]. 
The authors should replace
Level 3 [46]. 

Line 442
Varghese and colleagues [48].
Text: missing
Varghese et al. [48].......

Line 456
A study by Rosnan, R., et al.[49]. examined
The authors should replace
A study by Rosnan et al. [49] examined

Line 470
Komarov et al.[50]. explored
The authors should replace
Komarov et al. [50] explored

Comment 8
Figure 11
The Figure 11 must be accompanied on the same page as the Figure's title.

Line 509
al. [51] With Si3N4 
The authors should replace
al. [51]. With Si3N4 

Line 522
machineability
The authors should check the paper for typographical and spelling errors.

Line 529
Nohava et al.[53] investigated
The authors should replace
Nohava et al. [53] investigated

Line 542
Delete the space in the page end 

Line 554
was studied by Wei et al [56]. In
The authors should replace
was studied by Yongqiang et al. [56]. In

Line 556
by Yang et al.[58]. During the 
The authors should replace
by Yang et al. [58]. During the 

Line 581
Kumar and Patel,[59] investigated
The authors should replace
Kumar et al. [59] investigated

Line 587
Colombo et al [60] used laser
The authors should replace
Colombo-Pulgarin et al. [60] used laser

Comment 9
Table 1.
The Table 1 must be accompanied on the same page as the Table's title.

Line 627
hardened steel [94] . In
The authors should replace
hardened steel [94]. In

Line 671
Azinee and co-authors [97] coated
The authors should replace
Azinee et al. [97] coated

Lines 680 - 681
Uncoated surfaces are 1.154 m roughest and 0.42 m 
smoothest. The maximum and minimum Ra for coated substrates is 0.787 m and 0.251 m. 
The authors must check if the unit is m (perhaps μm?)

Line 707
process.. In the Co
The authors should replace
process. In the Co

Line 721
Suresh et al. [101] 
The authors should replace
Babu et al. [101] 

Line 739
Duran and colleagues [103] ,
The authors should replace
Duran et al. [103],

Line 742
Duran and colleagues [60].
The authors must check if the 60 is the right number.

Line 745
Devia et al. [104]. 
The authors should replace
Devia et al. [104] 

Line 753
Navarro-Devia et al.[105] discovered
The authors should replace
Navarro-Devia et al. [105] discovered

Line 766
Escobar et al.[106] deposited
The authors should replace
Escobar et al. [106] deposited

Line 774
Villarreal et al.[107] investigated
The authors should replace
Escobar et al.[107] investigated

Line 798
[HfN/VN]80 multilayered coatings
The authors should replace
[HfN/VN] 80 multilayered coatings

Line 807
Caicedo and coworkers [109], they
The authors should replace
Caicedo et al. [109], they

Line 818
Devia et al.[109] employed 
The authors must check if the 109 is the right number (or Devia).

Line 850
additives.[112]
The authors should replace
additives [112]

Line 877
Daroonparvar et al.[114]
The authors should replace
Daroonparvar et al. [114]

Line 887
he microstructure
The authors should check the paper for typographical and spelling errors.

Line 900
performance[116, 117].
The authors should replace
performance [116, 117].

Line 904
structure. [118, 199].
The authors should replace
structure [118, 199].

Line 905
Yamamoto et al.[120]
The authors should replace
Yamamoto et al. [120]

Line 917
Yamamoto et al.[121]
The authors should replace
Yamamoto et al. [121]

Line 930
.Nonetheless,
The authors should replace
. Nonetheless,

Line 965
Fukumoto et al.[127] investigated
The authors should replace
Fukumoto et al. [127] investigated

Line 969
Chang et al. investigated 
The number is missing.

Comment 10
Figure 20
The authors must explain what is in the right side of the Figure (550? - units?)

Line 1044
by Wanglin Chen et al [140], principally
The authors should replace
by Chen et al. [140], principally

Line 1106
Majid Abdoos et al [145]. report
The authors should replace
Majid Abdoos et al. [145] report

Lines 1111 - 1112
a scanning electron microscope was used to examine the prevalent wear pattern, partic- 
ularly on the rake face and cutting edge (SEM). 
The authors should replace
a scanning electron microscope (SEM) was used to examine the prevalent wear pattern, partic- 
ularly on the rake face and cutting edge. 

Line 1117
Andreas W. Nemetz et al 
The authors should replace
Nemetz et al. 

Lines 1123 - 1124
grain size of 1 m
with a 7 m
The authors must check if the "m" is right (maybe μm?)

Line 1135
Tlili et al [147].
The authors should replace
Tlili et al. [147].

Line 1183
tools. because this
The authors should replace
tools. Βecause this

Comment 11
The authors must insert a Table with the results of this study.
Better coating / Material

Comment 12
The authors must format the References according to the journal's instructions
References should be described as follows, depending on the type of work:
Journal Articles:
1. Author 1, A.B.; Author 2, C.D. Title of the article. Abbreviated Journal Name Year, Volume, page range.

Reviewer 3 Report

The authors present a comprehensive review of characterization and evaluation of optimally engineered coatings for cutting tools. However, this paper is not good enough for publishing. I suggest rejecting the manuscript and making significant changes considering the following comments and suggestions.

  1. The written English was not satisfactory. The authors wanted to write complex sentences but cannot express the content clearly and logically. English writing is not meet the requirement. It need be corrected by a native-speaker.
  2. The logicality of this paper should be improved, especially for the section of “Background”.
  3. Structure of the paper needs to be reorganized. Relations of the section 2 “coating design techniques”, section 3 “PVD coating”, and section 4 “PVD coating for end-milling cutting tools” were mixed. Relationship between these contents should be clarified and make them orderliness.
  4. The title of this paper is “characterization and evaluation of optimally engineered coatings for cutting tools——review”. However, there is a lack of content related to optimization.
  5. Quality of all pictures in the paper (especially, Figure. 4, Figure. 11, Figure 12, Figure 16, Figure 19, Figure 21, and Figure 22) should be improved so that readers can read the information clearly. Additionally, it should be noted that the use of error bars in Figure 13 needs to be modified.

Information exhibited in some pictures are confused with their title content. For example, Figure 7. (a) presented tool wear characteristic, while, the title read “Figure 7. (a) Diamond coated corner radius ……”. The same problem also appears in Figure 1, Figure 8, Figure 18, and Figure 20. It is suggested to check the whole paper to modify similar problems.

  1. Some expressions of coating materials are suggested to be amended. In the representation of coating materials, the number should be subscript. The representation in page 5/line 242 is acceptable. You had better check the full article and make sure that all the issues are modified.
  2. The format of some references needs to be modified in the references part. For example, reference [69] to reference [92] listed the name of journals. While, many references didn’t give the name of journals. The format of paper title in references are different, for example, reference [102] and reference [94].
  3. The content of page 2/line 45 to line 49 is not relevant to the content of the paper. This part is suggested to be deleted.
  4. The author need point out the current research challenges and possible solutions of the future clearly.

Round 2

Reviewer 1 Report

All corrections, suggestions and revisions made on the submitted comments found to be satisfactory. 

Reviewer 2 Report

Comment 1

The reviewer considers that 48 pages of review and 212 references is long enough for the journal.

Comment 2

Line 151

variability. but

The author must replace

variability. But

Line 170

cladding [31]. or to

The author must replace

cladding [31]. Οr to

The authors must check for typographical and spelling errors.

Comment 3

The authors must replace Fig. to Figure 

Comment 4

Table 1

The authors must explain the Coating Thickness (µm.) 

For example: For 2 papers, the Coating Thickness (µm.) are 1.2 - 6.3, 5, 0.2, 0.2, 0.1, 0.1

Comment 5

Lines 396 and 425

The authors must format the paper according to the journal's instructions.

Comment 6

Line 660

Suresh et al. [151] investigated

There is no Suresh in [151] (Babu et al. ?).

Comment 7 (previous Comment 12)

The authors must format the References according to the journal's instructions

References should be described as follows, depending on the type of work:

Journal Articles:

1. Author 1, A.B.; Author 2, C.D. Title of the article. Abbreviated Journal Name Year, Volume, page range.

Author's notes file: The references have been edited according to the journal style 

But, they did not make the corrections according to the journal's instructions (journal's name is missing).

Extended text editing for References needed. 

Reviewer 3 Report

 1. Quality of the pictures, such as Figure 3, Figure 4, Figure 6, and Figure 7, in this paper should be improved.

2. Please give the exact meaning of some symbol, such as the symbol of “Λ” in the line 739, in the paper.

3. Some expressions of coating materials still need to be amended. For example, the expressions of coating materials that appeared in line 878 to line 882 and that in line 949 to line 953 should be modified. In the representation of coating materials, the number should be subscript.
